# GRACE: A Language Model Framework for Explainable Inverse Reinforcement Learning

Silvia Sapora     Devon Hjelm     Alexander Toshev     Omar Attia     Bogdan Mazoure

## ABSTRACT

Inverse Reinforcement Learning aims to recover reward models from expert demonstrations, but traditional methods yield black-box models that are difficult to interpret and debug. In this work, we introduce GRACE (**G**enerating **R**ewards **A**s **C**od**E**), a method for using Large Language Models within an evolutionary search to reverse-engineer an interpretable, code-based reward function directly from expert trajectories. The resulting reward function is executable code that can be inspected and verified. We empirically validate GRACE on the MuJoCo, BabyAI and AndroidWorld benchmarks, where it efficiently learns highly accurate rewards, even in complex, multi-task settings. Further, we demonstrate that the resulting reward leads to strong policies, compared to both competitive Imitation Learning and online RL approaches with ground-truth rewards. Finally, we show that GRACE is able to build complex reward APIs in multi-task setups.

## 1 INTRODUCTION

The performance of modern Reinforcement Learning (RL) agents is determined by, among other factors, the quality of their reward function. Traditionally, reward functions are defined manually as part of the problem specification. In many real-world settings, however, environments are readily available while reward functions are absent and must be specified. Manually designing rewards is often impractical, error-prone, and does not scale, particularly in contemporary multi-task RL scenarios (Wilson et al., 2007; Teh et al., 2017; Parisotto et al., 2016).

A natural alternative is to automate reward specification by learning a reward model from data. The dominant paradigm here is Inverse Reinforcement Learning (IRL), which attempts to infer a reward model from observations of expert behavior (Ng & Russell, 2000; Christiano et al., 2017; Ziebart et al., 2008). In the era of Deep RL, approaches such as AIRL (Fu et al., 2018) represent rewards with deep neural networks. While effective, these reward functions are typically opaque black boxes, making them difficult to interpret or verify (Molnar, 2020). Moreover, IRL methods often require substantial amounts of data and can lead to inaccurate rewards (Sapora et al., 2024).

Recently, expressing reward models through code has emerged as a promising approach (Venuto et al., 2024a; Ma et al., 2023). Code is a particularly well-suited representation, as reward functions are often far simpler to express than the complex policies that maximize them (Ng & Russell, 2000; Cook, 1971; Godel, 1956). These approaches leverage code-generating Large Language Models (LLMs) and human-provided task descriptions or goal states to generate reward programs (Venuto et al., 2024a). Subsequently, the generated rewards are verified (Venuto et al., 2024a) or improved using the performance of a trained policy as feedback (Ma et al., 2023). However, this prior work has not investigated whether it is possible to recover a reward function purely from human demonstrations in an IRL-style setting, without utilizing any explicit task description or domain-specific design assumptions.

In this work, we address the question of how to efficiently infer rewards-as-code from expert demonstrations using LLMs. We propose an optimization procedure inspired by evolutionary search (Goldberg, 1989b; Eiben & Smith, 2003b; Salimans et al., 2017a; Romera-Paredes et al., 2024a; Novikov et al., 2025b), in which code LLMs iteratively introspect over demonstrations to generate and refine programs that serve as reward models. This perspective effectively revisits the IRL paradigm in the modern context of program synthesis with LLMs.

Our contributions are threefold:

- **Sample-Efficient Reward Recovery:** We demonstrate that code LLMs conditioned on expert demonstrations can produce highly accurate reward models that generalize well to held-out data. Crucially, GRACE is highly sample-efficient, recovering accurate rewards from relatively few demonstrations without requiring manual domain knowledge or human-in-the-loop guidance.

- **Well-Shaped Rewards:** We show that rewards learned by GRACE enable the training of strong policies across diverse domains, including the procedural *BabyAI*, continuous control in *MuJoCo*, and real-world device control in *AndroidWorld*. Empirical results indicate that GRACE matches or outperforms established IRL baselines (e.g., GAIL) and online RL with ground-truth, sparse rewards.

- **Interpretability and Modularity:** Unlike black-box neural networks, GRACE generates rewards as executable Python code, making them inherently interpretable and verifiable. Furthermore, in multi-task settings, our evolutionary search naturally emerges a modular library of reusable reward functions, enabling efficient generalization across tasks.

## 2 RELATED WORKS

**LLMs for Rewards** A common way to provide verification/reward signals in an automated fashion is to utilize Foundation Models. LLM-based feedback has been used directly by Zheng et al. (2023) to score a solution or to critique examples (Zankner et al., 2024). Comparing multiple outputs in a relative manner has been also explored by Wang et al. (2023). Note that such approaches use LLM in a zero shot fashion with additional prompting and potential additional examples. Hence, they can utilize only a small number of demonstrations at best. In addition to zero shot LLM application, it is also common to train reward models, either from human feedback (Ouyang et al., 2022) or from AI feedback (Klissarov et al., 2023; 2024). However, such approaches require training a reward model that isn't interpretable and often times require a larger number of examples.

**Code as Reward** As LLMs have emerged with powerful program synthesis capabilities (Chen et al., 2021; Austin et al., 2021; Li et al., 2023; Fried et al., 2022; Nijkamp et al., 2022) research has turned towards generating environments for training agents (Zala et al., 2024; Faldor et al., 2025) for various domains and complexities.

When it comes to rewards in particular, code-based verifiers use a language model to generate executable Python code based on a potentially private interface such as the environment's full state. Because early language models struggled to reliably generate syntactically correct code, the first code-based verifiers (Yu et al., 2023; Venuto et al., 2024b) implemented iterative re-prompting and fault-tolerance strategies. More recent approaches focus on progressively improving a syntactically correct yet suboptimal reward function, particularly by encouraging exploration (Romera-Paredes et al., 2024b; Novikov et al., 2025a). Other approaches such as Zhou et al. (2023); Dainese et al. (2024) use search in conjunction with self-reflection (Madaan et al., 2023) to provide feedback.

While closely related to our work, EUREKA (Ma et al., 2023) assumes access to the ground-truth reward signal to evaluate and evolve the generated code, while Reward-As-Code (Venuto et al., 2024b) depends on explicit task descriptions or goal states and a hand written pipeline. In contrast to these earlier works, our pipeline doesn't require any domain-specific adaptation and does not require the ground truth reward signal or a description of the task.

**Inverse Reinforcement Learning (IRL)** Early approaches infer a reward function that makes the expert's policy optimal over all alternatives (Ng & Russell, 2000). Subsequent works focused on learning policies directly, without explicit reward recovery (Abbeel & Ng, 2004), while incorporating entropy regularization (Ziebart et al., 2008) or leveraging convex formulations (Ratliff et al., 2006). More recently, deep IRL methods such as Generative Adversarial Imitation Learning (GAIL) (Ho & Ermon, 2016) have framed the problem as a distribution matching task using adversarial training. Adversarial Inverse Reinforcement Learning (AIRL) (Fu et al., 2018) further improves upon this by recovering disentangled and transferable reward functions. While related to our formulation, our representation (code) and our optimization strategy (evolutionary search) are fundamentally different, as GRACE benefits from implicit regularization through its symbolic reward representation.

## 3 BACKGROUND

**Reinforcement Learning** We consider a finite-horizon Markov Decision Process (MDP) (Puterman, 2014) parameterized by $\mathcal{M} = \langle \mathcal{S}, \mathcal{A}, T, r \rangle$ where $\mathcal{S}$, $\mathcal{A}$ are the state and action spaces, $T : \mathcal{S} \times \mathcal{A} \to \Delta(\mathcal{S})$ is the transition operator, and $r$ is a reward function. The agent's behavior is described by the policy $\pi : \mathcal{S} \to \Delta(\mathcal{A})$. Starting from a set of initial states $\mathcal{S}_0 \subset \mathcal{S}$, the agent takes the action $a \sim \pi(s)$ at $s$, receives a reward $r(s)$ and transitions into state $s' \sim T(s, a)$.

The performance of the agent is measured with expected cumulative per-timestep rewards, referred to as return:

$$J(\pi, r) = \mathbb{E}_{\tau \sim \pi, T}[\sum_{t=1}^{H} r(s_t)] \tag{1}$$

where $\tau$ are trajectory unrolls of horizon $H$ of the policy $\pi$ in $\mathcal{M}$. An optimal agent can be learned by maximizing Equation (1) via gradient descent with respect to the policy, also known as policy gradient (Sutton et al., 1999; Schulman et al., 2017).

**Inverse Reinforcement Learning** If the reward $r$ is not specified, it can be learned from demonstrations of an expert policy $\pi_E$. In particular, the classical IRL objective learns a reward whose optimal return is attained by the expert (Ng & Russell, 2000; Syed & Schapire, 2007):

$$\max_R \min_\pi J(\pi_E, r) - J(\pi, r) \tag{2}$$

More recent IRL approaches learn a discriminator that distinguishes between expert and non-expert demonstrations (Ho & Ermon, 2016; Swamy et al., 2021). The likelihood of the agent's data under the trained discriminator can be implicitly thought of as a reward. Modern approaches often frame this as a divergence minimization problem, matching the state-action visitation distributions of the learner and the expert. These approaches utilize gradient based methods to optimize their objectives.

**Evolutionary search** As an alternative for cases where the objective is not readily differentiable, gradient-free methods can be employed. One such method is evolutionary search, which maintains a set of candidate solutions (called a population) and applies variation operators to improve it (Salimans et al., 2017b; Eiben & Smith, 2003a; Goldberg, 1989a). These operators include mutation, where a hypothesis is partially modified, and recombination, where two hypotheses are combined to produce a new one. Each variation is evaluated using a fitness function, which measures the quality of a given hypothesis. Starting with an initial population, evolutionary search repeatedly applies these variation operators, replacing hypotheses with higher-fitness alternatives.

In this work, we focus on inferring reward functions, represented as Python code, from a set of demonstrations. While this setup is related to IRL, representing rewards as code prevents us from applying gradient-based methods commonly used in IRL. For this reason, we adopt evolutionary search as our optimization method.

## 4 METHOD

We propose GRACE, **G**enerating **R**ewards **A**s **C**od**E**, an interpretable IRL framework that generates a reward function as executable Python code. Initially, an LLM analyzes expert and random trajectories (Phase 1) and generates a preliminary set of reward programs. This initial set of reward functions is then iteratively improved through evolutionary search, where the LLM mutates the code based on low-fitness examples (Phase 2). The best-performing reward function is used to train an RL agent. This agent then explores the environment, and the new trajectories it generates are used to expand the dataset, revealing new edge cases or failure modes (Phase 3). This loop continues, with the expanded dataset from Phase 3 being used in the next iteration of Phase 1 and 2, progressively improving the reward function. Notably, prior to the initial analysis, the LLM can optionally perform a data cleaning step to identify and remove irrelevant states if the expert trajectories are known to be noisy or suboptimal. The overall process is illustrated in Figure 1 and detailed below and in Algorithm 1

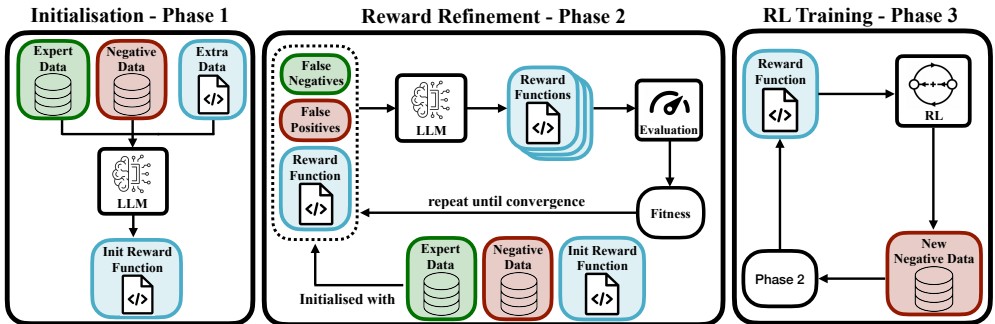

Figure 1: Overview of the GRACE framework. **(a)** The expert, negative and extra data (if any) is used to generate an initial set of possible reward functions. **(b)** The expert and negative states are used to mutate reward functions through an evolutionary procedure. The rewards are iteratively refined by feeding low-fitness examples to the reward. **(c)** An agent is trained with online RL using the converged reward; the data it sees during the training is added to $\mathcal{D}^-$ and used to further improve the reward.

**Phase 1: Initialisation** The initial reward code generation by GRACE is based on a set of demonstration trajectories $\mathcal{D}^+$ and a set of random trajectories $\mathcal{D}^-$. The former is generated using an expert policy or human demonstrations depending on the concrete setup, while the latter is produced by a random policy. Note that with a slight abuse of notation we will use $\mathcal{D}$ to denote interchangeably a set of trajectories as well the set of all states from these trajectories.

The language model is prompted with a random subset of $\mathcal{D}^+$ and, optionally, extra information available about the environment (e.g. its Python code or tool signature), to produce two artifacts:

**Initial rewards:** The LLM generates an initial set $\mathcal{R}^{\text{init}}$ of reward functions. Each function $r \in \mathcal{R}^{\text{init}}$ is represented as Python code:

```
def reward(state) -> float:
    <LLM produced code>
```

designed to assign high values to expert states $\mathcal{S}_e$ and low values to negative ones $\mathcal{S}_n$. This set of rewards is treated as the population in the subsequent evolution phase.

**(Optional) Data cleaning:** The LLM analyzes the states from expert demonstrations to identify the subset of expert states $\mathcal{S}_e \subseteq \mathcal{D}^+$ that solve the task - these are positive samples. All remaining states $\mathcal{S}_n = \{\mathcal{D}^+ \setminus \mathcal{S}_e\} \cup \mathcal{D}^-$ are treated as negative samples.

**Phase 2: Reward Refinement through Evolutionary Search** GRACE uses Evolutionary Search to discover rewards that effectively distinguish between expert and negative states. This is achieved by *mutating* the current reward population $\mathcal{R}$ using a code LLM and retaining rewards with high *fitness*.

The *fitness* $f$ of a reward function $r$ measures how well this function assigns large values to expert states and small values to negative states, akin to what would be expected from a meaningful reward. To do this, we adopt the standard AIRL (Fu et al., 2018) loss to ensure transferable rewards:

$$f(r) = \mathbb{E}_{s \sim \mathcal{S}_e}[\log D_r(s)] + \mathbb{E}_{s \sim \mathcal{S}_n}[\log(1 - D_r(s))] \tag{3}$$

where $D_r(s)$ is the discriminator parametrised by the reward function $r$. Following the AIRL formulation, the discriminator takes the form $D_r(s) = \frac{exp(r(s))}{exp(r(s)) + \pi(a|s)}$.

The *mutation* operator $m$ utilizes an LLM to improve the current reward code and correct failures. We prompt the LLM to generate a mutated reward $r'$ based on the current source code and specific failure cases:

$$m(r) = \text{LLM}(\text{source}(r), \text{context}, \text{prompt}) \tag{4}$$

Crucially, the context provided to the LLM enables targeted debugging. It includes:

- **Current Reward Source Code:** The Python implementation of the "parent" reward function $r$.

- **Wrong Examples:** A random sample of states $s_w$ with low fitness under reward $r$.

- **Reward Values:** The output value $r(s_w)$ by the current function for each low-fitness state.

- **(Optional) Extra environment information:** It is possible to include, if available, extra information about the environment such as (partial) source code or a text description.

- **(Optional) Debug information:** The LLM can define a $\texttt{debug}(s, D^+, D^-)$ function. The output of this function can then be included, in-context, in each mutation prompt. This allows the model to print custom intermediate variables, logic checks during execution or aggregate information over the expert and negative datasets.

We repeatedly apply the mutation operation to evolve the reward population $\mathcal{R}$. In each iteration, we sample a set of parent rewards $r \in \mathcal{R}$ based on a softmax distribution of their fitness, where the probability of selecting a specific reward is given by $\frac{\exp(f(r))}{\sum_{r' \in \mathcal{R}} \exp(f(r'))}$. We apply the mutation operator to these parents to generate a new batch of candidate rewards. To maintain a high-quality population, we retain only the top $N$ performing functions based on their fitness scores (from the combined set of current and newly mutated rewards). After $K$ iterations, we return the single reward function with the highest fitness $r^* = \arg\max_{r \in \mathcal{R}}\{f(r)\}$. This phase is detailed as function EVOSEARCH in Algorithm 1.

**Phase 3: Active Data Collection via Reinforcement Learning** The optimal reward $r^*$ above is obtained by inspecting existing demonstrations. To further improve the reward, we can collect further demonstrations by training a policy $\pi_{r^*}$ using the current optimal reward $r^*$; and use this policy to collect additional data $\mathcal{D}_{r^*}$.

In more detail, we employ PPO (Schulman et al., 2017) to train a policy in the environment of interest. As this process can be expensive, we use a predefined environment interaction budget $N$ instead of training to convergence. The new trajectories are added to the dataset of negative trajectories $D^-$, as they are likely to contain new edge cases the reward should consider. These are used to further refine the reward population as described in the preceding Sec. 4 (Phase 2). This phase is presented as function DATAEXPANDRL in Algorithm 1.

The final algorithm, presented in Algorithm 1, consists of repeatedly performing Evolutionary Search over reward population $\mathcal{R}$ followed by data expansion using RL-trained policy.

---

**Algorithm 1** GRACE: Generating Rewards As CodE

**Inputs:**
$\mathcal{D}^+$: expert trajectories, $\mathcal{D}^-$: random trajectories
$\mathcal{R}$: reward population
**Parameters:**
$P$: population size, $N$: RL budget
$M$: data augmentation steps

**procedure** GRACE($\mathcal{D}^+, \mathcal{D}^-$)
   $\mathcal{S}_e = \{s \in D^+\}, \mathcal{S}_n = \{s \in D^-\}$
   $\mathcal{R} = \{\text{LLM}(\mathcal{S}_e, \mathcal{S}_n, \text{reward\_prompt})\}$

   // Reward Refinement.
   **for** $i = 1 \ldots M$ **do**
      $\mathcal{R} \leftarrow \text{EVOSEARCH}(\mathcal{R}, \mathcal{S}_e, \mathcal{S}_n)$
      $\mathcal{D}_{r^*} \leftarrow \text{DATAEXPANDRL}(\mathcal{R})$
      $\mathcal{S}_n = \mathcal{D}_{r^*} \cup \mathcal{S}_n$
   **end for**
   **Return** $r^* = \arg\max_{r \in \mathcal{R}} f(r)$
**end procedure**

**function** EVOSEARCH($(\mathcal{R}, \mathcal{S}_e, \mathcal{S}_n)$)
   **for** $k = 1 \ldots K$ **do**
      $\mathcal{R}_{\text{new}} \leftarrow \emptyset$
      **for** $j = 1 \ldots P$ **do**
         Sample $r \sim \exp(f(r))$
         $r' \leftarrow m(r)$ // See Eq. 4
         $\mathcal{R}_{\text{new}} \leftarrow \mathcal{R}_{\text{new}} \cup \{r'\}$
      **end for**
      $\mathcal{R} \leftarrow \text{Top}_P(\mathcal{R} \cup \mathcal{R}_{\text{new}})$ // Keep best $P$ rewards
   **end for**
   **Return** $\mathcal{R}$
**end function**

**function** DATAEXPANDRL($\mathcal{R}$)
   $r^* \leftarrow \arg\max_{r \in \mathcal{R}} f(r)$
   Train $\pi_{r^*}$ with PPO under budget $N/M$
   Collect new trajectories $\mathcal{D}_{r^*}$
   **return** $\mathcal{D}_{r^*}$
**end function**

---

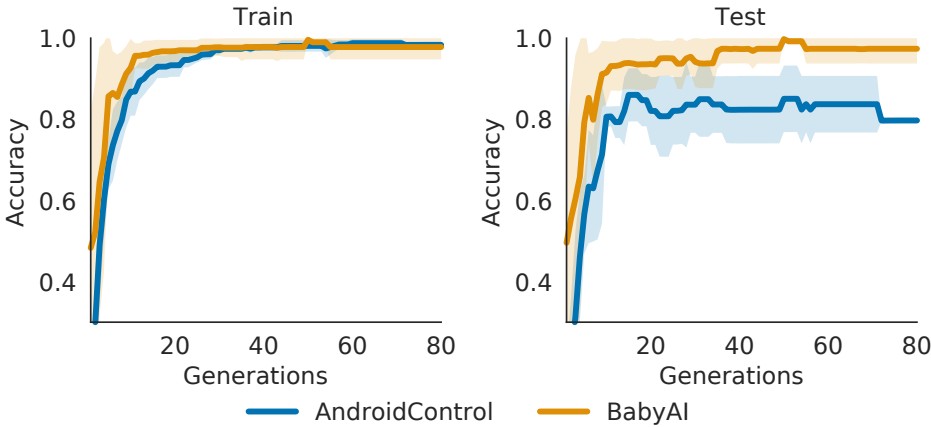

Figure 2: **Fitness vs Number of generations.** Evolution of train and test fitness across evolution generations, as defined by Algorithm 1, for *BabyAI* and *AndroidControl* (multi-level settings). For *BabyAI*, we provide 8 expert trajectories and 8 negative trajectories for each task. For *AndroidControl*, we provide 8-12 expert trajectories and $\sim$800 negative trajectories total. Shading is standard deviation across 3 seeds. For these experiments, no online data is added beyond the initial trajectories provided ($M = 1$).

**Additional reward shaping**   When the reward function offline performance on $\mathcal{D}$ doesn't translate to good online RL performance, we assume that the reward signal is poorly shaped, and additional refinement is required. In these cases, the LLM's info in Eq. 4 is augmented beyond low-fitness states to include full expert trajectories alongside the current reward value for each state. We then instruct the LLM to reshape the reward function, using expert trajectories as a reference, so that it provides a signal that increases monotonically.

## 5 EXPERIMENTS

We empirically evaluate GRACE with respect to its ability to generate rewards that lead to effective policy learning. Specifically, we aim to address the following questions:

- **Accuracy and Generalization**: Can GRACE recover correct rewards, and with how many expert samples?
- **Policy Learning Performance**: How does GRACE compare to other IRL methods or to online RL trained with ground-truth rewards?
- **Qualitative Properties:** How well-shaped are the rewards produced by GRACE?
- **Interpretability and Multi-Task Efficacy**: Does GRACE produce reusable reward APIs?

|  | PPO | GRACE w/ GPT-4o | GRACE w/ Qwen3-Coder-30B | GAIL w/ 200 traj | AIRL w/ 200 traj |
|---|---|---|---|---|---|
| **Hopper** | $2212 \pm 54$ | $2143 \pm 80$ | $2106 \pm 76$ | $2056 \pm 92$ | $2028 \pm 82$ |
| **Walker** | $2675 \pm 292$ | $2072 \pm 576$ | $2229 \pm 600$ | $1982 \pm 101$ | $2108 \pm 293$ |
| **Ant** | $6239 \pm 237$ | $5707 \pm 210$ | $6085 \pm 804$ | $5521 \pm 674$ | $4308 \pm 306$ |
| **Humanoid** | $6455 \pm 302$ | $5809 \pm 106$ | $5921 \pm 301$ | $6521 \pm 337$ | $6512 \pm 291$ |

Table 1: **MuJoCo Results** Average returns on 4 MuJoCo continuous control tasks. Average and standard deviation is reported across 5 different seeds. The total number of required LLM calls to recover a reward for each task averages at 2000 for both GPT-4o and Qwen3-Coder-30B.

## 5.1 EXPERIMENTAL SETUP

To evaluate GRACE, we conduct experiments in three distinct domains: the procedurally generated maze environment *BabyAI* (Chevalier-Boisvert et al., 2018), which tests reasoning and generalization; the physics simulator *MuJoCo* (Todorov et al., 2012), which ensures GRACE works on continuous, non-symbolic environments; and the Android-based UI simulator *AndroidWorld* (Rawles et al., 2024), to evaluate applicability to real-world, complex problems. Across all experiments, we don't provide GRACE with any in-context information about the environment (such as a description or source code) to ensure a fair comparison against baselines.

**BabyAI** Our *BabyAI* evaluation suite comprises 20 levels, including 3 custom levels designed to test zero-shot reasoning on tasks not present in public repositories, thereby mitigating concerns of data contamination. Expert demonstrations are generated using the `BabyAI-Bot` (Farama Foundation et al., 2025), which algorithmically solves *BabyAI* levels optimally. We extend the bot to support our custom levels as well. For each level, we gather approximately 500 expert trajectories. Another 500 negative trajectories are collected by running a randomly initialized agent in the environment. The training dataset consists of up to 16 trajectories, including both expert and negative examples. All remaining trajectories constitute the test set. For each dataset, we evolve the reward on the train trajectories and report both train and test fitness from Eq. (3). For this environment, we never augment the dataset with online trajectories ($M = 1$), as we establish it's not necessary to recover the optimal reward.

The state is represented by a $(h, w, 3)$ array. The state is fully observable, with the first channel containing information about the object type (with each integer corresponding to a different object, such as box, key, wall, or agent), the second channel contains information about the object's color and the third any extra information (e.g. agent direction, if is the door locked). The recovered reward function needs to operate on the state representation as a $(h, w, 3)$ array, but we also include a pixel image of the state in context for the LLM. For the classic GAIL baseline, we finetune `llava-onevision-qwen2-0.5b-ov-hf` on state images as the reward model. Given *BabyAI* is an environment that terminates on success, we only consider the last state of expert trajectories as a positive state, we do this for both GRACE and the GAIL baseline.

**MuJoCo** We conduct additional experiments on 4 tasks from the classical *MuJoCo* continuous control suite (Todorov et al., 2012): `Hopper, Walker, Ant, Humanoid`. These tasks demonstrate that GRACE also succeeds at reward design in continuous action and state spaces. We run all our MuJoCo experiments using the fully differentiable physics engine Brax (Freeman et al., 2021) to speed up learning. Unlike the *BabyAI* and Android experiments, in *MuJoCo* we augment the dataset 10 times ($M = 10$) with new trajectories coming from the learner policy. The reward is only updated if the fitness is low on the newly added trajectories.

**Android** To assess GRACE in a high-dimensional, real-world setting, we use the AndroidControl dataset (Rawles et al., 2023; Li et al., 2024), which provides a rich collection of complex, multi-step human interactions across standard Android applications. The state space includes both raw screen pixels and the corresponding XML view hierarchy. Both representations are included in-context in the mutation prompt, while the reward function only takes the XML representation as input.

From this dataset, we curate a subset of trajectories focused on the Clock application, where users successfully complete tasks such as "set a timer 6 hours from now". These serve as our positive examples. Negative samples are drawn from trajectories in other applications (e.g., Calculator, Calendar, Settings). Specifically, we select 12 expert demos for the "set timer" task, and 8 each for the stopwatch tasks ("pause stopwatch" and "run stopwatch"). We use 80% of trajectories in the train set and the remaining for the test set. For this environment, we never augment the dataset with online trajectories ($M = 1$), as we establish it's not necessary to recover the optimal reward. In this setting, we recover a multi-task reward and we give the option to the LLM to remove any states it judges as unnecessary to task completion.

**GRACE Parameters** All parameters used across our experiments can be found in Appendix A.4.

| Task | PPO | GAIL | GRACE |
|---|---|---|---|
| GoToRedBallNoDist | 1.00 | **1.00** | **1.00** |
| GoToRedBall | 1.00 | 0.35 | **1.00** |
| PickupDist | 0.31 | 0.15 | **0.32** |
| PickupLoc | 0.21 | 0.00 | **0.26** |
| GoToObj | 1.00 | 0.92 | **1.00** |
| OpenDoorColor | 1.00 | 0.98 | **1.00** |
| OpenTwoDoors | 1.00 | 0.37 | **1.00** |
| PlaceBetween (new) | 0.09 | 0.01 | **0.09** |
| OpenMatchingDoor (new) | 0.79 | 0.20 | **0.35** |
| Multi-task | 0.95 | 0.31 | **0.92** |

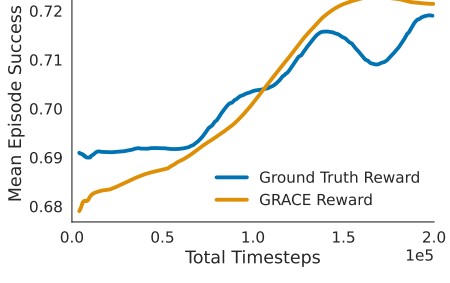

Table 2: **Success rates on selected BabyAI environments.** GRACE compared against PPO and GAIL. GRACE uses 8 expert trajectories per task, while GAIL uses 2000. Additional results are available in Table 3

Figure 3: **Training Curves for *Android-World* Clock Tasks.** Mean episode success over the 3 *AndroidWorld* clock tasks: ClockStopWatchPausedVerify, ClockStop-WatchRunning, and ClockTimerEntry.

## 5.2 ANALYSIS

**GRACE recovers rewards with high accuracy.** We first examine whether GRACE evolutionary search (Phase 2) can successfully recover the underlying task reward from demonstrations alone. We evaluate this in two settings using *BabyAI*: (i) a single-level setting, where the model infers a task-specific reward, and (ii) a more challenging multi-level setting, where GRACE must learn a single, general reward function conditioned on both state and a language goal.

In Figures 4 and 2, we show that the reward accuracy consistently reaches 1.0 across all BabyAI tasks in both single- and multi-level settings, as well as on AndroidControl. A fitness of 1.0 corresponds to assigning higher values to all expert states than to negative states.

We further ablate two aspects of the algorithm. First, we analyze sample efficiency by varying the number of expert and negative demonstrations. Results on BabyAI (Figure 4a) show non-trivial performance even with a single demonstration, with gradual improvement and perfect scores achieved using only eight expert trajectories. The number of negative trajectories also plays a role, though to a lesser degree: for example, accuracy of 0.95 is achieved with just a single negative trajectory, provided that sufficient expert trajectories are available (Figure 4b).

Finally, we assess the robustness and efficiency of the evolutionary process. As shown in Figure 2, in the multi-task setting GRACE reliably converges to a high-fitness reward function in fewer than 100 generations (i.e., evolutionary search steps) and no additional trajectories from the learner agent ($M = 1$), demonstrating the effectiveness of our LLM-driven refinement procedure.

**Comparison with Baselines**: To validate the quality of the inferred reward function, we compare against two approaches. First, we use PPO (Schulman et al., 2017) to optimise both the rewards recovered by GRACE as well as the groundtruth sparse reward. Clearly, the latter should serve as an oracle, while it does not benefit from dense rewards. As an IRL baseline, we compare against GAIL (Ho & Ermon, 2016). GAIL is trained with a large dataset of $2,000$ expert trajectories per task.

As shown in Tables 1 and 2, GRACE consistently matches or outperforms baselines across all tasks with less training data. On several tasks, GRACE matches Oracle PPO with ground-truth rewards, whereas GAIL completely fails. This demonstrates that the interpretable, code-based rewards from GRACE are practically effective, enabling successful downstream policy learning. To ensure a fair comparison, baseline and GRACE agents are trained using the same underlying PPO implementation, agent architecture and hyperparameters. Performance is measured by the final task success rate after 1e7 environment steps. Crucially, no extra information or environment code is provided in context to GRACE. Similarly, we evolve a separate reward function for each task in the AndroidControl dataset matching tasks present in the Clock *AndroidWorld* tasks: ClockStopWatchPausedVerify, ClockStopWatchRunning and ClockTimerEntry. The training curves for all tasks (averaged) are reported in Figure 3.

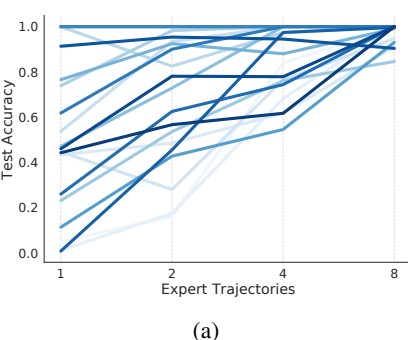
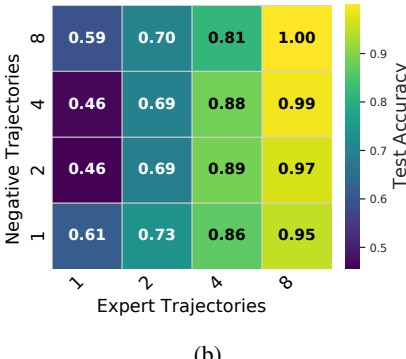

(a)  (b)

Figure 4: **Fitness vs Number of Expert Trajectories.** The accuracy is computed on test dataset after obtaining maximum fitness on training data with corresponding number of expert and negative training trajectories. (a) Performance on all 20 BabyAI tasks. (b) Aggregate accuracy across 20 BabyAI tasks.

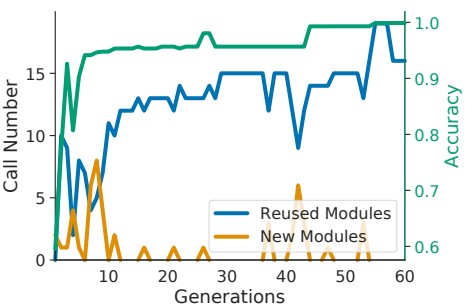
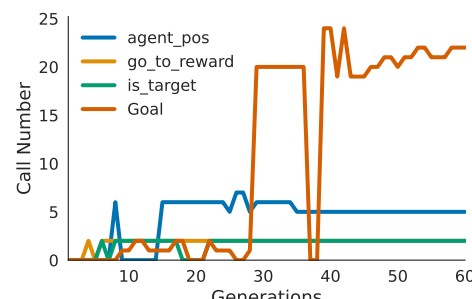

Figure 5: **Module and function reuse across generations** On the left, we show at each generation step the number of newly created modules and the number of existing and thus reused modules from prior rewards, contrasted with the accuracy in the reward population. On the right, we show number of times a module are being re-used, for a select set of modules.

**MuJoCo Performance** In our MuJoCo evaluation, GRACE leverages an LLM-generated debug function to analyze distributional differences between expert and negative trajectories, computing key statistics such as mean, standard deviation, percentiles, and Cohen's $d$ for each state feature. While this statistical analysis proves effective for lower-dimensional tasks, the method still struggles with the Humanoid environment. We hypothesize that this performance drop stems from the LLM's struggle to effectively parse and reason over the extensive numerical arrays required to represent high-dimensional states in a text-based format.

**GRACE generates well shaped rewards**: We demonstrate GRACE's ability to produce well-shaped rewards that accelerate learning. For challenging, long-horizon tasks like OpenTwoDoors, a correct but unshaped reward can lead to local optima where the agent gets stuck (Figure 6, "Gen 1"). By explicitly tasking the LLM to introduce shaping terms, GRACE refines the reward to provide a denser learning signal. As shown in Figure 6, this targeted shaping dramatically improves both the final performance and the speed of learning, allowing the agent to solve the task efficiently. This confirms that GRACE not only finds what the goal is but also learns how to guide an agent towards it.

**GRACE Code Reuse**: A key advantage of representing rewards as code is the natural emergence of reusable functions that collectively form a domain-specific reward library. We study this phenomenon in the multi-task *BabyAI* setting (Figure 5). In the early generations of evolutionary search, GRACE actively generates many new modules to explore alternative reward structures. After generation 10, the rate of new module creation drops sharply. At this point, GRACE shifts toward reusing the most effective, high-level modules it has already discovered.

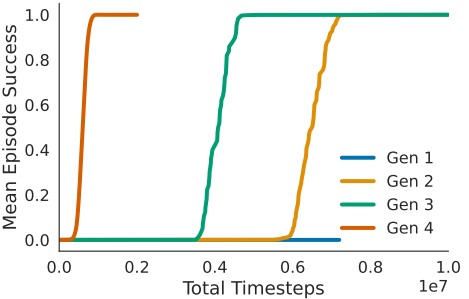 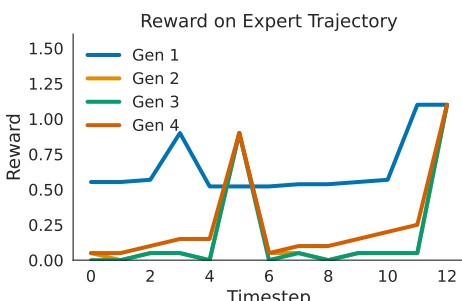

Figure 6: **Shaping** Using the default reward recovered by GRACE occasionally leads to failure in learning the correct behavior due to poor shaping. Through the targeted shaping in Phase 3, we significantly improve final performance and speed of learning.

To further illustrate this reuse, Figure 5 (right) shows call counts for a selected set of modules within the evolving reward API. For instance, the *Goal* module, which summarizes a set of goals, is initially used sparingly but becomes heavily invoked following a code refactor at generation 30. Likewise, the *agent_pos* function is reused at least five times after its introduction. These trends demonstrate that GRACE progressively builds a reward library that supports efficient multi-task generalization.

## 6 DISCUSSION

**Limitations**   A key limitation of GRACE is its limited scalability to high-dimensional state spaces for evolving reward functions. While code offers interpretability, neural networks are inherently better suited for processing high-dimensional observations (such as pixels), as they excel at learning distributed representations directly from raw sensory data, a task where symbolic feature extraction often proves brittle or infeasible.

**Conclusion**   We introduce GRACE, a novel framework that leverages LLMs within an evolutionary search to address the critical challenge of interpretability in IRL. Our empirical results demonstrate that by representing reward functions as executable code, we can move beyond the black-box models of traditional IRL and produce rewards that are transparent, verifiable, and effective in RL learning. We show that GRACE successfully recovers accurate and generalisable rewards from few expert trajectories, in stark contrast to deep IRL methods like AIRL. This sample efficiency suggests that the strong priors and reasoning capabilities of LLMs provide a powerful inductive bias. Furthermore, we demonstrate the framework's practical utility by applying it to the AndroidWorld environment, showing that GRACE can learn rewards for a variety of tasks with real-world applications directly from user interaction data.

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

# A    APPENDIX

## A.1    ADDITIONAL ONLINE RESULTS

| Task | PPO | GAIL
w/ 2000 trajs | GRACE
w/ 8 trajs |
|---|---|---|---|
| GoToRedBallNoDist | 1.00 | 1.00 | 1.00 |
| GoToRedBall | 1.00 | 0.35 | 1.00 |
| PickupDist | 0.31 | 0.15 | 0.32 |
| PickupLoc | 0.21 | 0.00 | 0.26 |
| GoToObj | 1.00 | 0.92 | 1.00 |
| OpenDoorColor | 1.00 | 0.98 | 1.00 |
| OpenTwoDoors | 1.00 | 0.37 | 1.00 |
| OpenRedDoor | 1.00 | 1.00 | 1.00 |
| GoToObjS4 | 1.00 | 1.00 | 1.00 |
| GoToRedBlueBall | 0.96 | 0.40 | 0.99 |
| GoToRedBallGrey | 0.97 | 0.77 | 0.99 |
| Pickup | 0.10 | 0.00 | 0.09 |
| Open | 0.30 | 0.18 | 0.22 |
| OpenRedBlueDoors | 1.00 | 0.96 | 0.98 |
| OpenDoorLoc | 0.39 | 0.40 | 1.00 |
| GoToLocalS8N7 | 0.64 | 0.39 | 0.97 |
| GoToDoor | 0.74 | 0.37 | 0.99 |
| SortColors (new) | 0.00 | 0.00 | 0.00 |
| PlaceBetween (new) | 0.09 | 0.01 | 0.09 |
| OpenMatchingDoor (new) | 0.79 | 0.20 | 0.35 |
| Multi-task | 0.95 | 0.31 | 0.92 |

Table 3: **Success rates on additional BabyAI environments**. The performance of our method, GRACE, is compared against two key baselines: PPO, trained on the ground-truth reward, and GAIL, trained using 2000 expert trajectories per task. GRACE's performance is evaluated with 8 expert trajectories per task to demonstrate its high sample efficiency. All values represent the final success rate at the end of training. We don't report GAIL's performance on 8 expert trajectories as it is near 0 for most tasks.

## A.2    EXTENDED DISCUSSION AND FUTURE WORK

GRACE's reliance on programmatic reward functions introduces several limitations, particularly when compared to traditional deep neural network based approaches. These limitations also point toward promising directions for future research.

**Input modality**    While generating rewards as code offers interpretability and sample efficiency, it struggles in domains where the reward depends on complex, high-dimensional perceptual inputs. Code is inherently symbolic and structured, making it less suited for interpreting raw sensory data like images or audio. For instance, creating a programmatic reward for a task like "navigate to the cat" is non-trivial, as "cat" is a difficult visual concept. NNs, in contrast, excel at learning features directly from this kind of data. Programmatic rewards can also be brittle: a small difference in the environment violating a hard-coded assumption could cause the reward logic to fail completely, whereas NNs often degrade more gracefully.

**Data Quantity**    GRACE demonstrates remarkable performance with very few demonstrations. This is a strength in data-scarce scenarios. However, it is a limitation when vast amounts of data are available. Deep IRL methods like GAIL are designed to scale with data and may uncover complex patterns from millions of demonstrations that would be difficult to capture in an explicit program. While GRACE's evolutionary search benefits from tight feedback on a small dataset, it is not clear how effectively it could leverage a massive, complex dataset.

**Failure Cases**   Although GRACE is highly sample-efficient, it can still fail. For example, in the BabyAI-OpenTwoDoors task, GRACE often proposed a reward that didn't take into account the order in which the doors were being opened. Similarly, in the new BabyAI-SortColors task, it would sometimes return a reward that only accounted for picking up and dropping both objects, without paying attention to where they were being dropped. While these errors can be easily fixed by providing a relevant negative trajectory or by treating all learner-generated states as negative trajectories, they highlight that GRACE can still misinterpret an agent's true intent based on expert demonstrations alone.

**Hybrid Approaches**   These limitations can be substantially mitigated by extending the GRACE framework to incorporate tool use, combining the strengths of both systems. The LLM could be granted access to a library of pre-trained models (e.g., object detectors, image classifiers, or segmentation models). The LLM's task would then shift from writing low-level image processing code to writing high-level logic that calls these tools and reasons over their outputs. A final direction involves generating hybrid reward functions that are part code and part neural network. The LLM could define the overall structure, logic, and shaping bonuses in code, but instantiate a small, learnable NN module for a specific, difficult-to-program component of the reward. This module could then be fine-tuned using the available demonstrations, creating a reward function that is both largely interpretable and capable of handling perceptual nuance. By exploring these hybrid approaches, future iterations of GRACE could retain the benefits of interpretability and sample efficiency while overcoming the inherent limitations of purely programmatic solutions in complex, perception-rich environments.

A.3 NEW BABYAI LEVELS

To evaluate the generalization and reasoning capabilities of GRACE and mitigate concerns of data contamination from pre-existing benchmarks, we designed three novel BabyAI levels.

**PlaceBetween**  The agent is placed in a single room with three distinct objects (e.g., a red ball, a green ball, and a blue ball). The instruction requires the agent to pick up a specific target object and place it on an empty cell that is strictly between the other two anchor objects. Success requires being on the same row or column as the two anchors, creating a straight line. This task moves beyond simple navigation, demanding that the agent understand the spatial relationship "between" and act upon a configuration of three separate entities.

**OpenMatchingDoor**  This level is designed to test indirect object identification and chained inference. The environment consists of a single room containing one key and multiple doors of different colors. The instruction is to "open the door matching the key". The agent cannot solve the task by simply parsing an object and color from the instruction. Instead, it must first locate the key, visually identify its color, and then find and open the door of the corresponding color. This task assesses the agent's ability to perform a simple chain of reasoning: find object A, infer a property from it, and then use that property to identify and interact with target object B.

**SortColors**  The environment consists of two rooms connected by a door, with a red ball in one room and a blue ball in the other. The instruction is a compound goal: "put the red ball in the right room and put the blue ball in the left room". To make the task non-trivial, the objects' initial positions are swapped relative to their goal locations. The agent must therefore execute a sequence of sub-tasks for each object: pick up the object, navigate to the other room, and drop it. This level tests the ability to decompose a complex language command and carry out a plan to satisfy multiple, distinct objectives.

## A.4 HYPERPARAMETERS

Table 4: Hyperparameters for Training BabyAI with PPO

| Parameter | Value |
|---|---|
| Base Model | llava-onevision-qwen2-0.5b-ov-hf |
| Gamma | 0.999 |
| Learning Rate | 3e-5 |
| Entropy Coef | 1e-5 |
| Num Envs | 10 |
| Num Steps | 64 |
| Episode Length | 100 |
| PPO Epochs | 2 |
| Num Minibatch | 6 |

Table 5: Hyperparameters for Training AndroidWorld

| Parameter | Value |
|---|---|
| Base Model | Qwen2.5-VL-3B-Instruct |
| LoRA Rank | 512 |
| LoRA Alpha | 32 |
| LoRA Dropout | 0.1 |
| Critic Hidden Size | 2048 |
| Critic Depth | 4 |
| Gamma | 0.999 |
| Learning Rate | 3e-5 |
| Entropy Coef | 0.0 |
| Num Envs | 16 |
| Num Steps | 16 |
| Episode Length | 20 |
| PPO Epochs | 2 |
| Num Minibatch | 2 |

Table 6: Hyperparameters for GRACE Evolution

| Parameter | Value |
|---|---|
| Population Size | 20 |
| Elite | 4 |
| Num Generations | 100 |
| Include expert trajectory chance | 0.25 |
| Incorrect state only chance | 0.5 |
| Expert state only chance | 0.75 |
| Model | gpt-4o |

## A.5 EVOLUTION EXAMPLES

```python
def _parse_colour_from_text(text: Optional[str]) -> Optional[int]:
    if text is None:
        return None

    colour_words: Dict[str, int] = {
        "red": 0,
        "green": 1,
        "blue": 2,
        "yellow": 3,          "purple": 3,
      "yellow":  4,
        "orange": 5,    # keep old mapping
      "grey":  5, # alias for the observed colour code in the trajectory
      "gray":  5,
        }
      lower = text.lower()
    for word, code in colour_words.items():
        if word in text.lower(): lower:
            return code
    return None

def _parse_goal_type(text: Optional[str]) -> str:
    if text is None:
        return "key"
    txt = text.lower()
    if "ball" in txt:
        return "ball"
      if "box" in txt:
    return "box"
        return "key"
```

Figure 7: **GRACE iteratively refines the initial BabyAI reward function (iteration 0) to handle unseen entities (iteration 10).** Using execution traces, the agent fixes its color code mistake and adds a new box entity.

```python
from __future__ import annotations

import re
from typing import Optional, Tuple

import numpy as np

COLOR2ID = {
    "red": 0,
    "green": 1,
    "blue": 2,
    "purple": 3,
    "yellow": 4,
    "grey": 5,
    "gray": 5,   # US spelling
}

OBJECT2ID = {
    "empty": 0,
    "wall": 1,
    "floor": 2,
    "door": 3,
    "key": 5,
    "ball": 6,
    "box": 8,
    "agent": 10,
}

DIR2VEC: dict[int, Tuple[int, int]]  ={
  0:  (1, 0), # south
  1:  (0, 1), # east
  2:  (-1, 0), # north
  3:  (0, -1), # west
}
```

```python
36  def _parse_goal(extra_info: str ) -> Tuple[int, Optional[int]]:
37      """Return *(object_id, colour_id)* parsed from *extra_info*."""
38      if not extra\_info:
39          raise ValueError("extra_info must specify the target, e.g. 'the red ball'.")
40
41      tokens = re.findall(r"[a-zA-Z]+", extra\_info.lower())
42      obj_id: Optional[int] = None
43      col_id: Optional[int] = None
44      for tok in tokens:
45          if obj_id is None and tok in OBJECT2ID:
46          if tok in COLOR2ID and col_id is None:
47              col_id = COLOR2ID[tok]
48          if tok in OBJECT2ID and obj_id is None:
49              obj_id = OBJECT2ID[tok]
50          if col_id is None and tok in COLOR2ID:
51              col_id = COLOR2ID[tok]
52      if obj_id is None:
53          raise ValueError(
54              f"Could not parse target object from extra_info='{extra_info}'."
55          )
56      return obj_id, col_id  # colour may be None (wild-card)
57
58
59  class Reward:
60      """Success when definition (single-step, dense reward):
61      100.0 âĂŞ the **first** cell in front of the agent is *either*
62      âĂŞ on / adjacent (according to the
63                      closest target object (Manhattan distance <= 1), OR
64      âĂŞ direction stored in the third observation channel) contains a
65      matching target has disappeared from the observable grid (picked up).
66
67      Shaping:  r = 1 / (1+d) with d the Manhattan distance to the closest
68      still-visible target, clipped at 0 object.
69      <1.0 âĂŞ shaping reward 1/(d+1) otherwise.
70      0.0 âĂŞ if either the agent or (a matching) target is out of view. not visible.
71
72      The implementation is modular so new goal
73      types can be handled by extending the OBJECT/COLOR lookup tables or by
74      replacing the *success predicate*.
75      """
76
77      SUCCESS_REWARD = 100.0
78      def __init__(self, extra\_info: Optional[str] str = None):
79          self.tgt_obj_id, self.tgt_col_id self._target_obj_id, self._target_colour_id =
80              _parse_goal(extra_info)
81      def __call__(self, state: np.ndarray) -> float:  # enable direct call
82      return self.reward_fn(state)
83
84      def reward_fn(self, state: np.ndarray) -> float:
85          """state:  (H, W, 3) """
86          agent_pos = self._find_agent(state)
87          if agent_pos is None:
88              return 0.0
89
90          # mask of all target objects still visible
91          tgt_mask = (state[:, :, 0] == self.tgt_obj_id) & (
92              state[:, :, 1] == self.tgt_col_id
93          )
94          if not tgt_mask.any():
95              # object gone -> picked up / carried
96              return self.SUCCESS_REWARD
97          # distance to the closest visible target
98          tgt_positions = np.argwhere(tgt_mask)
99          dists = np.abs(tgt_positions - agent_pos).sum(axis=1)
```

```
100          target_positions = self._find_targets(state)
101          if target_positions.size == 0:
102           # No matching target in view -> no shaping.
103           return 0.0
104          # -------------------------------------------
105          # Success predicate – target must be directly in front of the agent.
106          # -------------------------------------------
107          if self._is_target_in_front(agent_pos, state):
108           return 100.0
109          # -------------------------------------------
110          # Shaping:  inverse Manhattan distance (< 1.0) to the *nearest* target.
111          # -------------------------------------------
112          dists = np.abs(target_positions - agent_pos).sum(axis=1)
113        min_dist = int(dists.min())
114        if min_dist <= 1:
115          return self.SUCCESS_REWARD
116        return 1.0 / (1.0 + min_dist)
117      @staticmethod
118      def _find_agent(state: np.ndarray) -> Optional[np.ndarray]:
119          """Return (row, col) of """Locate the first agent
             pixel found, in the observation (row, col) or None.""" *None* if absent."""
120          locs = np.argwhere(state[:, :, 0] == OBJECT2ID["agent"])
121          if locs.size == 0:
122              return None
123          return locs[0]
124      def _find_targets(self, state: np.ndarray) -> np.ndarray:
125    """Return an (N, 2) array of row/col positions of matching targets."""
126    obj_mask = state[:, :, 0] == self._target_obj_id
127    if self._target_colour_id is not None:
128    col_mask = state[:, :, 1] == self._target_colour_id
129    mask = obj_mask & col_mask
130    else:
131    mask = obj_mask
132    return np.argwhere(mask)
133
134    def _is_target_in_front(self, agent_pos: np.ndarray, state: np.ndarray) -> bool:
135    """Return *True* iff the cell directly in front of the agent matches target."""
136    row, col = agent_pos
137    agent_dir = int(state[row, col, 2])
138    drow, dcol = DIR2VEC.get(agent_dir, (1, 0)) # default to south if unknown
139    f_row, f_col = row + drow, col + dcol
140
141    # Out of bounds – cannot be success.
142    if not (0 <= f_row < state.shape[0] and 0 <= f_col < state.shape[1]):
143    return False
144
145    # Check object id
146    if state[f_row, f_col, 0] != self._target_obj_id:
147    return False
148
149    # Check colour if colour was specified.
150    if (
151    self._target_colour_id is not None
152    and state[f_row, f_col, 1] != self._target_colour_id
153    ):
154    return False
155
156    return True
```

Figure 8: Example of code evolution across many generations.

## A.6 GENERATED REWARDS

### A.6.1 ANDROID: SET TIMER REWARD

```python
import json
import re
from typing import Optional, Tuple

# ------------------------------------------------------------
#              GENERIC & NORMALISATION HELPERS
# ------------------------------------------------------------

def _tab_selected(state: str, label: str) -> bool:
    pattern = (
        rf'"(content_description|text)"\s*:\s*"{label}"[^\n]*?"is_selected"\s*:\s*true'
    )
    return bool(re.search(pattern, state, re.I))

def _timer_tab_selected(state: str) -> bool:
    return _tab_selected(state, "Timer")

def _button_visible(state: str, label: str) -> bool:
    return bool(
        re.search(rf'"(content_description|text)"\s*:\s*"{label}"', state, re.I)
    )

def _is_timer_running(state: str) -> bool:
    return _button_visible(state, "Pause")

def _timer_keypad_mode(state: str) -> bool:
    return bool(re.search(r"\b\d{1,2}h\s*\d{1,2}m\s*\d{1,2}s\b", state))

# ------------------------------------------------------------
#                 TIMER / DURATION PARSING
# ------------------------------------------------------------

def _parse_requested_time(text: str) -> int:
    """Parses natural language (e.g. '5 minutes') into seconds."""
    text = text.replace("-", " ")
    hours = minutes = seconds = 0

    # Check explicit hours, minutes, seconds
    for patt, mult in (
        (r"(\d+)\s*hour", 3600),
        (r"(\d+)\s*minute", 60),
        (r"(\d+)\s*second", 1),
    ):
        m = re.search(patt, text, re.I)
        if m:
            val = int(m.group(1)) * mult
            if mult == 3600:
                hours = val // 3600
            elif mult == 60:
                minutes = val // 60
            else:
                seconds = val

    # Fallback: if "min" or raw number found
    if hours == minutes == seconds == 0:
        m = re.search(r"(\d+)\s*-?\s*min", text, re.I)
        if m:
            minutes = int(m.group(1))
        else:
            m = re.search(r"(\d+)", text)
            if m:
                minutes = int(m.group(1))

    total = hours * 3600 + minutes * 60 + seconds
    return total if total > 0 else 60

# ------------------------------------------------------------
#                    STATE EXTRACTION
# ------------------------------------------------------------

def _extract_timer_components(state: str) -> Optional[Tuple[int, int, int]]:
    """Extracts (HH, MM, SS) from the Android UI hierarchy state."""

    # Case 1: Text format like "5 minutes 0 seconds"
    m = re.search(r"(\d+)\s*minutes?\s*(\d+)\s*seconds", state, re.IGNORECASE)
    if m:
```

```python
 77            minutes = int(m.group(1))
 78            seconds = int(m.group(2))
 79            return (0, minutes, seconds)
 80
 81        # Case 2: Compact format "1h 05m 20s"
 82        m = re.search(r"(\d+)h\s*(\d+)m\s*(\d+)s", state, re.IGNORECASE)
 83        if m:
 84            hours = int(m.group(1))
 85            minutes = int(m.group(2))
 86            seconds = int(m.group(3))
 87            return (hours, minutes, seconds)
 88
 89        # Case 3: "MM:SS" format (excluding standard time like 12:30 PM)
 90        for mm_match in re.finditer(r"(\d{1,2}):(\d{2})(?!\s*[AaPp][Mm])", state):
 91            mm, ss = int(mm_match.group(1)), int(mm_match.group(2))
 92            if not (0 <= ss < 60):
 93                continue
 94            context = state[mm_match.end() : mm_match.end() + 80].lower()
 95            if "minute" in context or "timer" in context or "remaining" in context:
 96                return (0, mm, ss)
 97
 98        # Case 4: Digging through UI node hierarchy if Timer tab is active
 99        if not _timer_tab_selected(state):
100            return None
101
102        tokens = re.findall(r'"text"\s*:\s*"([^"]+)"', state)
103        tokens = [t.strip() for t in tokens]
104
105        # Look for HH:MM:SS pattern in tokens
106        for i in range(len(tokens) - 4):
107            if (
108                re.fullmatch(r"\d{1,2}", tokens[i])
109                and tokens[i + 1] == ":"
110                and re.fullmatch(r"\d{2}", tokens[i + 2])
111                and tokens[i + 3] == ":"
112                and re.fullmatch(r"\d{2}", tokens[i + 4])
113            ):
114                h = int(tokens[i])
115                m_val = int(tokens[i + 2])
116                s = int(tokens[i + 4])
117                if 0 <= m_val < 60 and 0 <= s < 60:
118                    return (h, m_val, s)
119
120        # Look for MM:SS pattern in tokens
121        for i in range(len(tokens) - 2):
122            if (
123                re.fullmatch(r"\d{1,2}", tokens[i])
124                and tokens[i + 1] == ":"
125                and re.fullmatch(r"\d{2}", tokens[i + 2])
126            ):
127                m_val = int(tokens[i])
128                s_val = int(tokens[i + 2])
129                if 0 <= s_val < 60:
130                    return (0, m_val, s_val)
131
132        return None
133
134 def _safe_json_dumps(obj) -> str:
135     try:
136         return json.dumps(obj, ensure_ascii=False)
137     except Exception:
138         return json.dumps({"error": "debug-serialization failed"})
139
140 # ------------------------------------------------------------
141 #                       REWARD CLASS
142 # ------------------------------------------------------------
143
144 class Reward:
145     """Dense reward function specifically for setting Google Clock Timers."""
146
147     def __init__(self, extra_info: Optional[str] = None):
148         self.raw_instr: str = extra_info or ""
149         self.instruction: str = self.raw_instr.lower()
150
151         self.task_type = "set_timer"
152
153         # Goal parsing
154         self.goal_seconds = _parse_requested_time(self.instruction)
155         rem = self.goal_seconds % 3600
156         self.goal_hms = (self.goal_seconds // 3600, rem // 60, rem % 60)
157
```

```
158            self.goal_achieved = False
159            self._t = 0
160
161        def reward_fn(self, state: str) -> float:
162            self._t += 1
163            if self.goal_achieved:
164                return 100.0
165
166            # Since we are only doing set_timer:
167            return self._reward_timer(state)
168
169        def debug_fn(self, state: str) -> str:
170            dbg = {
171                "step": self._t,
172                "task_type": self.task_type,
173                "goal_achieved": self.goal_achieved,
174                "goal_hms": self.goal_hms,
175            }
176            return _safe_json_dumps(dbg)
177
178        def _reward_timer(self, state: str) -> float:
179            reward = 0.0
180
181            # Check if we are on the correct tab
182            if _timer_tab_selected(state):
183                reward += 0.2
184
185            # Extract current digits entered into the timer
186            current_val = _extract_timer_components(state)
187            if current_val is None:
188                return min(reward, 0.99)
189
190            cur_hh, cur_mm, cur_ss = current_val
191
192            # Normalize to digit strings for comparison (strip leading zeros)
193            current_digit_string = f"{cur_hh:02d}{cur_mm:02d}{cur_ss:02d}".lstrip("0")
194            if current_digit_string == "":
195                current_digit_string = "0"
196
197            goal_digit_string = f"{self.goal_hms[0]:02d}{self.goal_hms[1]:02d}{self.goal_hms[2]:02d}".lstrip("0")
198            if goal_digit_string == "":
199                goal_digit_string = "0"
200
201            # Exact match check
202            if current_digit_string == goal_digit_string:
203                # We treat exact digit entry as success (even if not started yet)
204                # or if the timer is already running with those values.
205                return 100.0
206
207            # Partial match reward (dense reward for typing correct digits)
208            matching_digits = 0
209            for i in range(0, min(len(current_digit_string), len(goal_digit_string))):
210                if goal_digit_string[i] == current_digit_string[i]:
211                    matching_digits += 1
212                else:
213                    # Stop counting as soon as a mismatch occurs
214                    break
215
216            # Scale reward based on progress
217            if len(goal_digit_string) > 0:
218                reward += (matching_digits / len(goal_digit_string)) * 0.7
219
220            return min(reward, 0.99)
```

Listing 1: Android Control "Set Timer" Generated Reward.

### A.6.2 MuJoCo: Hopper

```
1   # imports
2   import jax
3   import jax.numpy as jnp
4
5   # reward function
6   @jax.jit
7   def reward_fn(state):
8       reward = 0.0
9
10      # Strong bonus for negative values in features 0, 1 (expert states are typically negative)
11      reward += jnp.minimum(0.0, state[0]) * 2.0  # Bonus for negative feature 0
12      reward += jnp.minimum(0.0, state[1]) * 2.0  # Bonus for negative feature 1
13
14      # Strong penalty for positive values in features 0, 1 (learner states often positive)
15      reward -= jnp.maximum(0.0, state[0]) * 2.0  # Penalty for positive feature 0
16      reward -= jnp.maximum(0.0, state[1]) * 2.0  # Penalty for positive feature 1
17
18      # Bonus for positive values in feature 5 (expert states are typically positive)
19      reward += jnp.maximum(0.0, state[5]) * 3.0  # Bonus for positive feature 5
20
21      # Penalty for negative values in feature 5 (learner states often negative)
22      reward -= jnp.maximum(0.0, -state[5]) * 1.0  # Penalty for negative feature 5
23
24      # Additional penalties for extreme positive values in key features
25      # Features 2,3,4,6,7,10: penalize large positive values (learner indicator)
26      reward -= jnp.maximum(0.0, state[2]) * 1.0
27      reward -= jnp.maximum(0.0, state[3]) * 1.0
28      reward -= jnp.maximum(0.0, state[4]) * 1.0
29      reward -= jnp.maximum(0.0, state[6]) * 1.0
30      reward -= jnp.maximum(0.0, state[7]) * 1.0
31      reward -= jnp.maximum(0.0, state[10]) * 1.0
32
33      # Additional penalty for large negative values in features 0,1 (to avoid over-rewarding)
34      reward -= jnp.maximum(0.0, -state[0]) * 0.5
35      reward -= jnp.maximum(0.0, -state[1]) * 0.5
36
37      return reward
```

Listing 2: MuJoCo Hopper Generated Reward.

## A.7 PROMPTS

---

**Goal Identification Prompt For Data Cleaning**

```
Given this reward code:  {reward_code}
```

**Trajectory:**
```
{trajectory}
```

```
Please analyze the state sequence and the agent's instruction.
Identify the index of the goal state.  The state indices are 1-based.
```

**OUTPUT FORMAT:**
```
Answer in a json format as follows:
'reasoning':  Explain your reasoning for choosing the goal state(s).
'goal_state_indexes':  A list of integers representing the 1-based
index of the goal state(s), or -1 if no goal state is present.
```

---

Prompt 1: The prompt for identifying the goal state(s) within a trajectory.

**LLM Initial Reward Generation**

You are an ML engineer writing reward functions for RL training.
Given a trajectory with marked goal states, create a Python reward
function that can reproduce this behavior.

**Requirements:**

- Write self-contained Python 3.9 code
- Make the function generic enough to handle variations
  (different positions, orientations, etc.)
- Design for modularity - you might extend this reward later to
  handle multiple goal types
- Aim to give high rewards for expert states and low rewards for
  all other states

**Environment Details:**
{env_code}, {import_instructions}, {state_description}

**Trajectories**
{expert_trajectories}

**Key Instructions:**

- Analyze the trajectory to understand what constitutes success
- Identify intermediate progress that should be rewarded
- Create utility functions for reusable reward components

The code will be written to a file and then imported.
**OUTPUT FORMAT:**
Answer in a json format as follows:
'reasoning':  Given the reason for your answer
'reward_class_code':  Code for the Reward function class in the
format:

```
# imports
<imports_here>
# utils functions
<utils functions here>
# reward function
class Reward:
    def __init__(self, extra_info=None):
      <code_here>

    def reward_fn(self, state):
      <code_here>

    def debug_fn(self, state):
      <code_here>
```

The Reward class will be initialized with the extra_info argument.
Describe in the comments of the class the behaviour you are trying to
reproduce.
reward_fn and debug_fn receive only state as argument.  The debug_fn
should return a string that will be printed and shown to you after
calling reward_fn on each state.  You can print internal class
properties to help you debug the function.  Extract any needed
information from the state or store it in the class.  The Reward
class will be re-initialised at the beginning of each episode.

Prompt 2: Prompt to generate the initial set of rewards

**Evolution Mutation Prompt**

You are an ML engineer writing reward functions for RL training. Given a trajectory with marked goal states, create a Python reward function that can reproduce this behavior.

**Requirements:**

- Write self-contained Python 3.9 code
- Aim to give high rewards for expert states and low rewards for all other states
- Make the function generic enough to handle variations (different positions, orientations, etc.)
- Design for modularity – you might extend this reward later to handle multiple goal types

**Original Reward Code:**
{{code}}

{{import_message}}
{{state_description}}

--
**CRITICAL: Incorrect Trajectories**
The reward function is not performing well on the following trajectories. It either assigned a high reward to a negative states or assigned low reward to an expert state. The predicted rewards for each step are shown.
Change the reward function to fix these errors.

**Key Instructions:**

- Create utility functions for reusable reward components
- Implement goal switching logic using extra_info to determine which reward function to use
- Reuse existing utilities where possible
- Make sure the logic you write generalises to variations in extra_info

{incorrect_trajectories}

{expert_traj_str}
--

Now, provide the mutated version of the reward function that addresses these errors.

**OUTPUT FORMAT:**
Answer in a json format as follows:
'reasoning': Briefly explain the corrective change you made.
'reward_class_code': Code for the Reward function class in the format:
# Reward format and extra info as above

Prompt 3: The prompt used for evolutionary mutation, providing feedback on incorrect trajectories.

