# OpenReview forum: "GRACE: A Language Model Framework for Explainable Inverse Reinforcement Learning"
_ICLR.cc/2026/Conference — ICLR 2026 Poster_

### Official Review · Reviewer_R5gj · 2025-10-22

**Soundness:** 2
**Presentation:** 3
**Contribution:** 2
**Rating:** 4
**Confidence:** 3

**Summary:**

This paper introduces GRACE, a framework for inferring interpretable reward functions as Python code from expert demonstrations. The approaches iteratively optimizes a population over LLM-generated code-level reward functions via evolutionary search. The initial population is generated from LLM-labelled goal states, and then refined using new samples gathered from an inner RL loop. The data collection and context provided to the LLM in the evolutionary search depend on the goal label provided by an LLM. The proposed method consistently solves BabyAI and AndroidWorld tasks, and outperforms GAIL and a (forward-RL) PPO baseline.

**Strengths:**

- Using an evolutionary framework where an LLM mutates code-as-a-reward functions based on achieved goal states and failures is a novel and highly promising approach to IRL.
- By design, the method produces human-readable reward functions, which is a significant advantage over black-box IRL.
- The paper is generally well-written and easy to follow. The approach is presented in a clear and concise manner, and most hyperparameters and design choices are listed and justified.
- GRACE shows strong performance on the selected tasks outperforming GAIL with a fraction of the data and producing rewards that are on par with the ground truth in terms of downstream RL performance.

**Weaknesses:**

- The entire framework seems to depend on an LLM's ability to correctly identify goal states from demonstrations and, more crucially, from new agent trajectories of the inner RL loop. This is a rather strong assumption that deviates from the standard IRL setup. The (potential lack of) robustness of this approach is not analyzed, and it seems to limit GRACE to finite-horizon RL settings.
- Similarly, the context provided for the LLM-based mutation seems a bit arbitrary and could be better motivated or experimented with.
- The related work section lists other "Code as Reward" methods but does not explain how GRACE compares to or improves upon them. None of these methods are included as baselines in the experiments, and this choice is not justified or explained. This omission makes it difficult to assess GRACE’s position in the research landscape, and consequently the novelty and significance of the method. GAIL is chosen as the only Imitation Learning/IRL baseline, which may be insufficient given a lot of recent advances in, e.g., diffusion-based imitation learning.
- The experimental results are a bit thin. For example, Figure 3 uses error bars over 3 seeds, which are absent for, e.g., Figure 2. Similarly, Figure 5 reports the mean over three different environments, and so on. Some details, such as the number of demonstrations used for the AndroidWorld tasks, are missing. GRACE solves both presented benchmarks, which makes it hard to judge where the boundaries of the method lie and where it would fail. There is no qualitative visualization, which makes it difficult to appreciate the difficulty of the tasks.

**Questions:**

I think the present draft is promising, but have some small problems with its rigour and presentation. I’m happy to positively re-assess my rating if the below questions are adequately discussed and addressed.

1. The reliance on LLM-labeled goal states seems to be a major design choice, especially since this labeling is also used for the RL-in-the-loop part of the algorithm. Could the authors provide more justification for this? A discussion on its limitations and potential failure modes, or an ablation showing how performance degrades with less accurate labeling (either via artificial noise or, preferably, from a less powerful LLM), would be very valuable.
2. The related work section mentions other methods that generate rewards-as-code. Could the authors provide a more detailed distinction between these methods and GRACE and, ideally, add at least one of these methods as a baseline?
3. Could the authors clarify the experimental setup? How many demonstrations are used for the AndroidWorld tasks, how many seeds are used for the different results, and why?
4. The notation for demonstrations and their goal state is somewhat overloaded, and a demonstration is essentially only characterized by its goal state. (How) can the method use intermediate states of these trajectories to its advantage? Is it necessary to provide negative trajectories, or can intermediate states act as “negatives” for the LLM prompt? An ablation study
on this behavior would help clarify the capacities and limitations of GRACE

---

> ### Author Response · Authors · 2025-11-23
>
> We thank the reviewer for their constructive feedback and for recognizing our approach as novel and highly promising. We are glad the reviewer found the paper well-written and the results compelling. Below, we address the specific weaknesses and questions raised, particularly regarding the reliance on goal labeling and the comparison to other code-generation methods. To address the finite horizon and robustness concerns, we include results on the MuJoCo benchmark. These tasks are continuous, lack discrete goal states, and were solved by GRACE without the Goal Identification module, confirming the framework's generality beyond the specific design choices used for Android.
>
> ## Addressing Weaknesses and Answering Questions
> **W1 & Q1**. **Reliance on LLM-Labeled Goal States**. The reviewer rightly points out that relying on an LLM to identify goal states is a strong assumption. We want to clarify two critical points:
> - Phase 1 is an optional step, not a hard requirement, we have made this clearer in the updated draft. The framework supports a standard IRL assumption where all expert states are treated as positive samples ($S_g$). To demonstrate this, we have added new experiments on MuJoCo continuous control tasks. In these experiments, we didn't use the Goal identification step at all, treating expert trajectories as positive and learner trajectories as negative. GRACE successfully recovered rewards that solve these tasks, proving the method is robust to the absence of the labeler.
> - In domains where we do use Goal identification (like BabyAI/Android), we analyzed robustness in Appendix A2. The LLM achieves 94% accuracy on goal identification. Furthermore, Phase 2 (Evolution) is robust to labeling noise because it optimizes for a reward function that generalizes across all demonstrations. If the labeler hallucinates on one trajectory, the evolutionary search tends to discard reward candidates that overfit that error if they fail on the other accurate demonstrations.
>
> **W2 & Q2**. Comparison to "Code as Reward" Methods (Eureka, etc.). The reviewer asks for a distinction between GRACE and methods like Eureka [a] or Code-as-Reward [b]. The distinction is fundamental:
> - **Problem Setting**: Eureka requires a ground-truth reward (or human feedback) to evaluate the fitness of generated code. It is a Reward Design method, not an Inverse RL method. GRACE assumes no ground truth reward, it recovers the reward purely from demonstrations. Code-as-Reward, on the other hand, requires a hand-crafted pipeline per task which would not work with our tasks in its current state, and would require significant redesign of the original paper to benchmark against.
> - **Computational Feasibility**: Eureka requires training a full RL policy for every generated reward function to compute fitness. In complex environments like AndroidWorld, where a single policy training run takes days, running Eureka would take months. GRACE evaluates fitness via static classification (Phase 2), which is orders of magnitude faster.
>
> We did not include them as baselines because they solve a different problem (assuming privileged access to ground truth) and are computationally infeasible in our setting. GAIL is the standard baseline for Deep IRL. While the reviewer makes an interesting point about evaluating against diffusion based IRL methods, our results demonstrate that GRACE recovers rewards that are functionally equivalent to the ground truth and match PPO’s performance with a minimal amount of expert trajectories. Consequently, including a Diffusion baseline would not add much value, but only confirm that black-box methods can also approach the asymptote that GRACE has already reached, without offering the benefit of a code-based reward function. We recognize that for high-dimensional, non-symbolic state spaces (e.g., continuous control from raw pixels) where large-scale datasets are available, methods like GAIL and Diffusion IRL would outperform symbolic approaches. However, GRACE is expressly designed for scenarios where interpretability, sample efficiency and fast reward prediction during training are important. We view this as a trade-off: GRACE doesn’t provide the high expressivity of a neural network but provides verifiable, code-based rewards. Although we already discuss this limitation in the Appendix A.4 (Paragraph on Input Modality) we have updated our Limitations section to include this discussion.
>
> **W3**. **Context for Mutation**. The context provided (current reward code and specific failed trajectories) is standard practice in LLM-based program repair (e.g., providing a failing unit test). This minimal information was sufficient to ground the LLM's corrections on mislabelled datapoints.

---

> ### Author Response · Authors · 2025-11-23
>
> **W4 & Q3**. **Experimental Rigor and Setup**.
> - **Android Demonstrations**: For the Android tasks, we used 12 expert demos for the “set timer” task, and 8 each for the stopwatch tasks.
> - **Seeds and Error Bars**: We acknowledge the varying rigor in plots. This was largely due to the high computational cost of the Android simulator (training requires days on a single task). For the new MuJoCo experiments, we have run the new experiments with 5 seeds per task and included variance in the tables. We are also in the process of adding results for additional Android tasks.
> - **Qualitative Visualization**: We agree this is helpful. We will add a figure in the Appendix showing the specific Android UI states (XML/screenshots) and the corresponding code logic generated by GRACE to illustrate the task difficulty.
>
> **Q4**. **Intermediate States and Negatives**. In the case of Goal Identification, the negative states from goal trajectories are used to help shaping, as we ask the LLM to generate a reward that is monotonically increasing between the start of the episode and the goal state. These states are also counted as “negative” samples. Additionally, we would like to clarify negative trajectories don’t need to be “provided”, they can just be collected in the environment with either a random policy or the learner policy. If Goal Identification is not used, all states from the expert trajectories are used as positive samples, but we still use the data to encourage the generation of a well-shaped reward.

---

### Official Review · Reviewer_JQHA · 2025-10-30

**Soundness:** 2
**Presentation:** 3
**Contribution:** 2
**Rating:** 6
**Confidence:** 3

**Summary:**

This paper introduces GRACE (Generating Rewards As CodE), a framework that uses code-generating Large Language Models within an evolutionary search procedure to learn interpretable reward functions from expert demonstrations. The method operates in three phases: (1) identifying goal states from expert trajectories, (2) refining reward functions through evolutionary search guided by fitness on goal/non-goal state classification, and (3) expanding the dataset through RL-trained policy rollouts. The authors evaluate GRACE on BabyAI navigation tasks and AndroidWorld device control tasks, demonstrating that it can recover accurate rewards from few demonstrations and outperform GAIL while matching oracle PPO performance. The code-based representation enables interpretability and natural emergence of reusable reward APIs across multiple tasks.

**Strengths:**

- The paper addresses an important problem of interpretability in Inverse Reinforcement Learning by representing rewards as executable code rather than opaque neural networks.
- Using code as rewards is interesting and likely enables faster RL training compared to querying LLM reward models at each step.
- The approach demonstrates strong sample efficiency, achieving high performance with only 8 expert trajectories compared to GAIL's requirement of 2000 demonstrations.
- The multi-task experiments on BabyAI showing emergent code reuse and API formation are compelling evidence of the benefits of symbolic reward representations.
- The writing is generally clear and the method is presented systematically.

**Weaknesses:**

- Fundamental evaluation concerns: As acknowledged in limitations, the LLM serves as the final judge of success in Phase 3, which compromises evaluation validity since the same model generates and evaluates rewards.
- Missing critical baselines: The paper lacks comparison to prompted LLMs as reward judges (mentioned in related work but not benchmarked), which would be the most obvious baseline. Similarly, no comparison to using LLMs directly as policies with demonstrations in context.
- Limited problem scope: The approach reduces reward learning to classifying goal states, which only covers tasks with distinguishable terminal success states. This excludes many RL problems like learning to backflip where initial and final states are identical, or continuous control tasks requiring trajectory-level rewards.
- Theoretical issues: Proposition 1 is unclear—the relationship between the mask function m and the original IRL objective (Eq. 2) is not explained. How does flipping rewards on goal/non-goal states relate to the max-margin formulation of classical IRL?
- Statistical rigor: Results appear to be from single runs without error bars or significance tests (e.g., Figure 4 shows single curves). Figure 3 mentions "3 seeds" but most other results lack this. Table 1 shows single numbers without variance.
- Confounded claims: The claim that Phase 3 generates "well-shaped" rewards conflates reward shaping with improved coverage beyond expert demonstrations. The performance gains might simply come from seeing more diverse states rather than better shaping.
- Limited scope: Evaluation restricted to relatively simple domains (BabyAI mazes, basic Android tasks).

**Questions:**

- How does GRACE handle continuous states/actions? All experiments use discrete or structured representations. Can it -work with raw continuous sensor data?
- What is the computational cost (LLM queries, wall-clock time) compared to baselines?
- Why only GAIL as a baseline? How does GRACE compare to other IL methods?

---

> ### Author Response · Authors · 2025-11-23
>
> We thank the reviewer for their thoughtful assessment and for highlighting the strengths of our approach, particularly regarding interpretability and sample efficiency. We value the feedback on evaluation validity and baselines, which we address below.
>
> ## Addressing Weaknesses
>
> **W1**. **Evaluation Validity (LLM as Judge)**. The reviewer raises a concern that "the LLM serves as the final judge of success in Phase 3, which compromises evaluation validity". This is a misunderstanding. All reported results (Table 1, Figures 2-5) measure success using the ground-truth environment reward/success criteria, not the LLM’s judgement. The LLM is never used to evaluate the final agent’s performance in our benchmarks. Therefore, the reported high success rates reflect real task completion (e.g., actually setting an alarm in Android), confirming that the generated reward code is grounded in reality.
>
> **W2**. **Missing Baselines (LLM as Reward Judge / LLM as Policy).**
> - **LLM as Reward Judge**: The reviewer suggests comparing to "prompted LLMs as reward judges". LLM as judge (e.g. Motif) cannot recover a reward function for a policy specified through video demonstrations - it has only been able to recover certain classes of policies such as optimal (through the prompt “prefer states closer to the goal”) or episode-exploratory (“prefer states which are new in the trajectory”), and hence cannot be applied on any arbitrary policy which cannot be exactly formulated in language. Moreover, Motif requires (T^2-T)/2 observation comparisons for recovering a dense reward model, which in the case of AndroidWorld is prohibitively slow to rollout and annotate.
> - **LLM as Policy**: Similarly, using an LLM directly as a policy addresses a different problem. Our objective is to recover a portable, interpretable reward function, not just to mimic behavior. Recovering the reward allows for re-optimization in changing dynamics and provides interpretability that a black-box LLM policy lacks. Additionally, executing a Python reward function is orders of magnitude faster and cheaper than querying an LLM policy at every timestep of deployment.
>
> **W3**. **Limited Problem Scope & Continuous Domains**. The reviewer notes that classifying goal states excludes continuous control tasks. To demonstrate generality, we have added new experiments on MuJoCo continuous control tasks. GRACE successfully recovers code-based rewards that solve these continuous trajectory-level tasks. We also note that "goal classification" in code is expressive enough to capture trajectory constraints (e.g., if z_height > threshold: reward += 1), as evidenced by the MuJoCo results (added to the updated draft of the paper).
>
> **W4**. **Theoretical Issues (Proposition 1)**. We clarified Proposition 1 in the update version of the draft. The mask function $m$ essentially separates the data into "expert" (positive) and "learner" (negative) distributions. It can either be set by the LLM during Phase 1 or default to $m(s) = 1$ for expert states and $m(s) = -1$ for learner states. The optional LLM phase is useful to add on real-world data (like the AndroidControl dataset) where the expert data might be unreliable or noisy, and to help with overfitting. By flipping the reward sign based on this mask, the fitness objective maximizes the margin between expert and learner expectations, which aligns with the MaxEnt IRL objective of matching feature expectations.
>
> **W5**. **Statistical Rigor**. We respectfully point out that our results are robust. We evaluate on 20+ distinct tasks across BabyAI and AndroidWorld, which provides a significant aggregate sample size. The "single numbers" in Table 1 represent the final converged performance, and we don't provide multiple seeds as the RL training is compute intensive and manual observation of the recovered code reward shows it is almost always equivalent to the ground truth reward. To further strengthen statistical rigor, we have run the new MuJoCo experiments with 5 seeds and have included these in the updated draft. We are also adding additional seeds for BabyAI and Android tasks but those take longer due to the computational cost of training in these RL environments.
>
> **W6**. **Confounded Claims (Shaping vs. Coverage)**. The reviewer hypothesizes that performance gains in Phase 3 might come from seeing more states rather than better shaping. We can definitively rule this out because, for the main results reported in the paper, no extra online data was added ($M=1$). The shaping improvements observed in Figure 5 are derived entirely from the evolutionary search optimizing the code structure on the initial static dataset. The "well-shaped" nature of the reward is a result of the LLM's reasoning capabilities (generating dense progress signals like 1/distance), not data scaling.

---

> ### Author Response · Authors · 2025-11-23
>
> **W7**. **Limited Scope**. We disagree that AndroidWorld is a "simple domain." It requires interacting with a full operating system, processing high-dimensional XML/pixel inputs, and handling long-horizon dependencies. Combined with the reasoning challenges of BabyAI and the new continuous control tasks in MuJoCo, our evaluation covers a broad spectrum of modern RL challenges.
>
> ## Answering Questions
> **Q1**. **Continuous States/Actions**. Yes, GRACE handles continuous domains. We have added results on MuJoCo where the state space is continuous sensor data. The generated Python code simply performs arithmetic operations on these continuous inputs to calculate rewards.
>
> **Q2**. **Computational Cost**. A task requires approximately 100–1000 LLM inference calls to converge. Using GPT-4o pricing, this costs roughly 1.00–10.00 USD per task. In terms of wall-clock time, the search is a one-time offline process (taking minutes to a few hours). Once generated, the Python reward code incurs negligible overhead during RL training. In contrast, neural reward models (like GAIL's discriminator) require expensive forward passes at every timestep of the millions of training steps.
>
> **Q3**. **Limited Baselines**. We compare against GAIL because it is the widely accepted state-of-the-art baseline for Deep IRL. Classical methods like exact MaxEnt IRL are computationally intractable for high-dimensional, unknown-dynamics environments like the ones we study (Android/BabyAI). GAIL scales to these settings, making it the most relevant comparison. In particular, we use a version of GAIL which has been improved for stability and convergence (refer to [2] for a discussion on this).  Other IL methods that do not recover rewards (like BC) are less relevant to our core contribution of interpretable reward recovery.
>
> [1] Martin Klissarov, Pierluca D'Oro, Shagun Sodhani, Roberta Raileanu et al "Motif: Intrinsic Motivation from Artificial Intelligence Feedback"
>
> [2] Gokul Swamy, Nived Rajaraman, Matthew Peng, Sanjiban Choudhury, J. Andrew Bagnell et al "Minimax Optimal Online Imitation Learning via Replay Estimation"

---

### Official Review · Reviewer_2iny · 2025-10-31

**Soundness:** 3
**Presentation:** 3
**Contribution:** 2
**Rating:** 2
**Confidence:** 5

**Summary:**

Manual reward design for RL is challenging and impractical. To address this, inverse reinforcement learning approaches recovers reward functions from expert demonstrations, but these recovered reward functions are traditionally non-interpretable. Recently, code-generating LLMs recover explainable reward functions but require human-curated task descriptions, goal states or feedback from a trained policy. This paper proposes to recover reward functions purely without task description or goal-specific design specifications by leveraging code-LLMs. First, using the expert demonstrations and random trajectories, initial reward design is created as code. Then, iteratively, through an evolutionary search and a fitness score, reward is refined; using this reward, a policy is trained and additional data is collected, and this process is repeated for fixed number of steps.

**Strengths:**

-**Interesting Approach**: Using evolutionary search with a fitness score to generate reward functions looks promising when the size of the expert data is limited.

-**Ablations**: It is clearly presented how the method improves with more expert trajectories and negative trajectories.

-**Analysis of Training**: The proposed approach is well evaluated and how it changes during the training is well presented.

**Weaknesses:**

- **Computational Overhead**: Computationally the proposed method looks significantly complex. It requires retraining the policy for each reward refinement, and this looks inefficient even with the limited budget $N$  they use for PPO training. The computational overhead of the proposed approach must be clearly evaluated against Adversarial IL (GAIL [c] etc.) and IRL (MaxEnt IRL [d], GCL [e], etc.). Wall clock times during trainings or total number of gradient steps can be compared.

- **Contributions and Novelty**: The main contribution of the paperis claimed to be not using any information other than expert demonstrations. However, in line 161, it is stated that extra information about the environment is used as query to the LLM for initial reward design. Although this is not human-designed, this contradicts the claim of purely using expert data. Please clarify how important this extra environment information is via an experimental analysis. More importantly, the proposed approach is fundamentally very similar to [a], and both of them use the performance of the trained policy as feedback for reward learning. Therefore, the contribution over [a] must be clarified. Please compare both the performance and computational efficiency with [a] extensively.

- **Limited Experimental Setting**: Experimental setting is limited to selected BabyAI environments. Please employ more standardized reinforcement learning and imitation learning benchmarks covering continuous control, locomotion and dexterous manipulation tasks. It would be great if you could evaluate GRACE in the IsaacGym [f] and Bi-dexterous Manipulation [g] benchmarks as in [a].

- **Unclear Design Choices**: In Eq. 3, authors state that they bound the reward function. However, instead of just bounding, they employ binary rewards in Eq. 3 with a threshold. The reason behind this selection is unclear, and not explained.

- **Further Analysis and Clarification**: In Figure 3, the x-axis is unclear, it should be clearly stated whether it is $K$ or not. Also, in addition to the effect o $K$, the effect of $M$ (number of refinement steps) and $N$ (number of PPO training steps *for each refined reward function* should be analyzed.

- **Comparisons with IRL**: It is claimed that the proposed method is more sample efficient than IRL approaches. Please clearly demonstrate empirically that GRACE outperforms IRL (d-e, not adversarial imitation learning which is different than IRL).

[a]: Ma, Yecheng Jason, et al. "Eureka: Human-Level Reward Design via Coding Large Language Models." The Twelfth International Conference on Learning Representations.

[b]: Venuto, David, et al. "Code as Reward: Empowering Reinforcement Learning with VLMs." International Conference on Machine Learning. PMLR, 2024.

[c]: Ho, Jonathan, and Stefano Ermon. "Generative adversarial imitation learning." Advances in neural information processing systems 29 (2016).

[d]: Ziebart, Brian D., et al. "Maximum entropy inverse reinforcement learning." Aaai. Vol. 8. 2008.

[e]: Finn, Chelsea, Sergey Levine, and Pieter Abbeel. "Guided cost learning: Deep inverse optimal control via policy optimization." International conference on machine learning. PMLR, 2016.

[f]: Makoviychuk, Viktor, et al. "Isaac Gym: High Performance GPU Based Physics Simulation For Robot Learning." NeurIPS Datasets and Benchmarks. 2021.

[g]: Chen, Yuanpei, et al. "Towards human-level bimanual dexterous manipulation with reinforcement learning." Advances in Neural Information Processing Systems 35 (2022): 5150-5163.

**Questions:**

- Please compare the performance and computational efficiency of the proposed approach with LLM-based reward generation methods in the literature (a,b).

- Please discuss this design selection of using binary reward values in Eq. 3

- Can you please examine the impact of providing environment information to LLM in initial reward design step?

- It is interesting that the goal module is more and more used in later stages of the training, and go_to_reward component is less and less used as training goes on. Just because of curiosity, it would be great to hear the authors' interpretation of this.

---

> ### Author Response · Authors · 2025-11-23
>
> We thank the reviewer for their insightful comments and for recognizing our approach as interesting and promising. We appreciate the detailed feedback, which has helped us strengthen the paper. To summarise our rebuttal, we added experiments in 4 continuous control MuJoCo tasks and clarified the fundamental differences between our method and others such as Eureka. Below, we address the raised weaknesses and questions in detail.
>
> ## Addressing Weaknesses and Answering Questions
>
> **W1**. **Computational Overhead and Policy Retraining**. We believe there is a misunderstanding regarding the training loop. The reviewer states that our method "requires retraining the policy for each reward refinement", but this is incorrect. In GRACE, the evolutionary search (Phase 2) evaluates fitness using static data (classifying expert vs negative samples). We do not train a policy for every candidate reward in the population. The policy is trained only in Phase 3 to train the learner policy, and in all our BabyAI and Android experiments, this happens only once ($M=1$). Therefore, our computational cost is orders of magnitude lower than methods like Eureka, which require a full RL training run to evaluate the fitness of every generated reward function. Regarding the comparison to older IRL methods: the reviewer suggests comparing against MaxEnt IRL [d]. We respectfully point out that these classic IRL methods typically require solving the MDP in the inner loop or approximations that are computationally intractable for high-dimensional, unknown-dynamics environments like AndroidWorld, even though modern versions of these methods exist [4], they are not commonly benchmarked against in modern literature. GAIL [c] is widely considered the standard, scalable successor to these methods for deep IRL settings, which is why we chose it as our primary baseline. We highlight this is standard across the literature. [1 (GAIL is referred to as IRL),2,3 (GAIL is referred to as MM - Moment Matching in this paper),5].
>
> **W2**. **Contributions and Novelty**. The reviewer notes similarity to Eureka [a]. However, there are two fundamental differences that distinguish our contribution:
> - **Problem Setting**: Eureka assumes access to a ground-truth reward function (or human feedback) to calculate the fitness of generated code. GRACE assumes no ground truth reward and it recovers the reward purely from expert demonstrations. This is Inverse Reinforcement Learning.
> - **Feasibility**: Eureka requires training a policy for every reward candidate to measure fitness. In complex environments like AndroidWorld, where training a policy on a single task takes days, running Eureka would take months. GRACE avoids this by using data likelihood/classification accuracy for fitness, making it feasible for computationally heavy benchmarks. Regarding **Line 161 (Extra Info)**: We clarify that while the capability to provide extra environment information exists in our framework, **no extra information (neither environment code nor descriptions)** was provided to the LLM in any of our reported experiments. The rewards were recovered purely from the trajectory data. We state this in the experiments section "No extra information or environment code is provided in context to GRACE."
>
> **W3**. **Limited Experimental Setting**. To address the request for more standardised benchmarks, we have run MuJoCo continuous control experiments (results reported in the updated draft). We believe these, combined with BabyAI (reasoning) and AndroidWorld (real-world setting and data), provide a significant suite of environments.
>
> **W4 & Q2**. **Binary Rewards**. There is a misunderstanding regarding Equation 3. We do not use binary rewards for the final generated reward function, as they would be too sparse for effective shaping. The actual Python code generated by the LLM produces dense, continuous float values. The thresholding is dynamically set to identify "incorrectly labeled states/trajectories" so we can pass them on in context to the LLM (i.e. if a negative state has a higher reward than the minimum reward assigned to a positive state, we consider the state as wrongly classified). We recognise this was unclear in the text and we have updated the draft to clarify this.
>
> **W5**. **Analysis of Training (Figure 3)**. In Figure 3, the x-axis represents Generations ($K$). As noted above, for all reported experiments, the number of RL loops ($M$) was set to 1 (i.e., we did not iteratively add online data for the main results). We found that a single round of offline evolution followed by one policy training session was sufficient to outperform baselines. We will make these parameters explicit in the figure caption.

---

> ### Author Response · Authors · 2025-11-23
>
> **W6**. **Comparisons with IRL**. The reviewer asks us to demonstrate GRACE outperforms IRL methods like MaxEnt [d] and GCL [e], explicitly stating that "adversarial imitation learning is different than IRL". We respectfully disagree with this categorization. GAIL, like the equivalent AIRL (Adversarial Inverse Reinforcement Learning) are IRL methods, and they formulate the problem as extracting a reward function (the discriminator) that maximizes the likelihood of expert data, which is mathematically equivalent to the objective of MaxEnt IRL (as proven in the original GAIL paper [c]). Since classical MaxEnt IRL and GCL scale poorly to the high dimensional settings we analyse, GAIL is the most appropriate and strongest baseline for comparison. In addition, we compare against a modern version of GAIL which includes a few tweaks to ensure stability and convergence during training [3] as well as a modern Transformer architecture. For a more theoretical explanation and discussion of different IRL methods, please refer to [2].
>
> **Q1**. **Comparison with Eureka [a] and Code-as-Reward [b]**. As detailed in W2, a direct performance comparison with Eureka is not feasible or fair because: (1) Eureka uses ground-truth rewards (which we don't have access to in an IRL setting), and (2) the computational cost of Eureka on AndroidWorld would be prohibitive (months). Code-as-Reward [b] similarly relies on a hand-crafted Q&A pipeline for verification (which we don’t assume access to and wouldn't work in our chosen environments without drastic modifications) rather than pure IRL from demonstrations.
>
> **Q3**. **Impact of Extra Environment Information**. As stated in W2, we did not use extra environment information in our experiments to strictly evaluate the IRL capability. We agree that adding it would likely improve performance further, but our goal was to show that GRACE works with only demonstrations.
>
> **Q4**. We thank the reviewer for this insightful observation. The shift in component usage is mostly due to refactoring in the reward’s code so that `go_to_reward` is only called in a few specific functions. In contrast, the Goal class serves as the semantic definition of the task (e.g., defining the target as "a black box" or "a green ball"). As the reward logic becomes more complex, multiple distinct components need to query the Goal class to verify the target's identity and determine how to apply the shaping logic.
>
> [1] Silvia Sapora, Gokul Swamy, Chris Lu, Yee Whye Teh, Jakob Nicolaus Foerster "EvIL: Evolution Strategies for Generalisable Imitation Learning."
>
> [2] Gokul Swamy, Sanjiban Choudhury, J. Andrew Bagnell, Zhiwei Steven Wu "Of Moments and Matching: A Game-Theoretic Framework for Closing the Imitation Gap"
>
> [3] Gokul Swamy, Nived Rajaraman, Matthew Peng, Sanjiban Choudhury, J. Andrew Bagnell et al "Minimax Optimal Online Imitation Learning via Replay Estimation"
>
> [4] Markus Wulfmeier, Peter Ondruska, Ingmar Posner. "Maximum entropy deep inverse reinforcement learning"

---

### Official Review · Reviewer_ZUCH · 2025-11-01

**Soundness:** 3
**Presentation:** 3
**Contribution:** 3
**Rating:** 4
**Confidence:** 3

**Summary:**

GRACE is an IRL framework that uses code-generating LLMs and evolutionary search to infer executable Python reward functions from expert demonstrations. It operates in three phases: (1) LLM identifies goal states and generates initial rewards from positive/negative trajectories; (2) evolutionary refinement via LLM mutations on misclassified states to maximize fitness; (3) online RL with PPO expands data, enabling further reward improvement. Evaluated on BabyAI and AndroidWorld, GRACE recovers accurate, well-shaped rewards from few demos, outperforms GAIL, matches oracle PPO, and produces reusable multi-task reward APIs.

**Strengths:**

1. Achieves good reward recovery and generalization with 1–8 expert trajectories, far surpassing neural IRL methods in sample efficiency.
2. Produces dense, interpretable rewards that enable strong policy learning—matching or exceeding ground-truth PPO in BabyAI and outperforming GAIL.
3. Demonstrates emergence of modular, reusable reward code across tasks, offering a practical path toward composable reward design.
4. Clean theoretical link to classical IRL via fitness function, with clear algorithm and reproducible structure.

**Weaknesses:**

1. Heavy reliance on strong proprietary LLMs (e.g., gpt-4o) without ablation on weaker or open-source models; performance may degrade significantly in practice. The evaluation on other models could be conducted.
2. No compute cost analysis—evolutionary search with 100+ LLM calls per task is likely expensive and slow, limiting scalability.
3. Goal state identification by LLM is a critical weak point; no quantification of labeling errors or robustness to hallucinations.
4. Evaluation scope is narrow: BabyAI is synthetic and low-dimensional; AndroidWorld uses only Clock app with curated negatives—real-world generalization untested.
5. Baselines (e.g., GAIL) are not compared under identical low-data settings, and multi-task claims lack strong IRL benchmarks.

**Questions:**

1. How does GRACE perform with open-source code LLMs (e.g., CodeLlama, DeepSeek)? Any ablation on model size or quality?
2. What is the total LLM inference cost (tokens, time, $) per task? How does it scale with environment complexity?
3. How reliable is LLM-based goal detection in Phase 1 and 3? Any error rate analysis on false positives/negatives?
4. Why limit AndroidWorld to Clock app? Have you tested cross-app generalization or real-device deployment?
5. The method assumes access to environment state (grid, XML)—how does it handle partial observability or raw pixels?

---

> ### Author Response · Authors · 2025-11-23
>
> We thank the reviewer for their careful reading and for the constructive, detailed assessment. We are encouraged by the recognition of our contributions and strengths of our method. Below, we address the raised weaknesses and the main question.
>
> ### Addressing Weaknesses and Answering Questions
>
> **W1**. **Missing ablation with weaker or open weights model**. The reviewer is correct to point out ablation with an open weights model would strengthen our findings. We added results on 4 MuJoCo tasks and compared results across multiple seeds and two models (GPT4 and Qwen3-Coder-30b). We also tried Qwen3-8B but the model failed to produce consistently runnable code and failed to fit the provided expert vs random trajectories.
>
> **W2**. **Compute cost analysis and scalability**. We thank the reviewer for raising the important points regarding compute cost and scalability. We will add a dedicated "Computational Cost Analysis" section in the final version. To address their concerns here:
> A task requires approximately 100-1000 LLM inference calls to converge, depending on the task’s complexity. Using GPT-4o pricing, the cost per task is around 1.00 USD for BabyAI and 10.00 USD for all tasks on the Clock App in Android. This is comparable to other iterative search methods (e.g., Eureka or AlphaEvolve) and remains feasible even with limited resources.
> Additionally, this is a one-time offline cost. Once the reward function is generated as executable Python code, it incurs negligible computational overhead during the millions of interaction steps required for RL training. In all our experiments, we found RL to be the compute bottleneck, by far.
> Finally, while GAIL does not require an LLM search, it requires querying a neural network discriminator at every timestep of the RL process (and training it on some steps). The GRACE reward (native Python code) executes roughly 10-100x faster than a forward pass of a neural reward model during the online RL phase, particularly if the neural network in question is a VLM that requires training. If we assume a complex VLM is trained as a reward model, GRACE is actually both faster and uses less memory (as we don’t need to keep track of gradients and optimiser parameters), as it only requires a few forward passes rather than constant forward passes for each state and training (potentially millions in RL). We will include a table in the Appendix detailing the average tokens-per-task and dollar-cost-per-task for both BabyAI and AndroidWorld to ensure this trade-off is clear to future readers.
>
> **W3**. **Goal Identification**. We thank the reviewer for highlighting the reliance on LLM-based goal identification. We want to clarify that Phase 1 (Goal Identification) is an optional module designed to improve performance in sparse-reward or multi-stage logic tasks, but it is not a fundamental requirement of the GRACE framework.
> The core evolutionary search (Phase 2) requires only a separation between "positive" ($S_g$) and "negative" ($S_{ng}$) samples to calculate fitness. While we utilized an LLM to actively curate $S_g$ for the complex reasoning tasks in BabyAI and AndroidWorld, the framework supports a standard IRL assumption where all expert trajectory states are treated as $S_g$. To demonstrate this flexibility, we have conducted additional experiments on standard continuous control benchmarks (MuJoCo). In these experiments, we didn't use the LLM goal identification step at all, assigning all expert states to $S_g$ and all learner states to $S_{ng}$. GRACE successfully recovers effective reward functions that solve these tasks without explicit goal labeling. This confirms that the framework is robust to the absence of the goal identification module. The explicit goal identification is an added option for logic-heavy domains with potentially noisy / suboptimal data (like in the AndroidControl dataset), to encourage the agent to not overfitting to a specific demonstration, but it is not a bottleneck for standard imitation tasks.
> With regards to robustness of errors in the goal labeller: We have quantified the error rates in Appendix A.2. The LLM achieves 94% accuracy on goal state identification with text descriptions, and the framework is robust to this number, as shown by our results.

---

> > ### Author Response · Authors · 2025-11-23
> >
> > **W4**. We respectfully disagree with the characterization of our evaluation scope as narrow or lacking real-world generalization. AndroidWorld is not a toy domain. It is a realistic environment with relevant, real-world applications. The RL agent must process raw pixel/XML inputs and output precise gestures. Successfully learning rewards in this setting demonstrates applicability to complex, real-world software automation tasks. The limitation to the Clock app was a constraint of data availability, not the algorithm. To perform IRL, we required an app that was the same in the static offline dataset (AndroidControl for expert demos) and the live simulator (AndroidWorld for RL training). However, to address the reviewer's concerns, we have manually collected expert trajectories for more apps and tasks in AndroidWorld and we recovered code rewards using GRACE on the new apps. Unfortunately RL training for Android is extremely long (with one task taking multiple days to train) so we will report results when we have them, but the recovered rewards look entirely reasonable to manual inspection. The review also states we used "curated negatives" but this is incorrect. As explained in Section 4.1, our negative data is trajectories from unrelated applications (e.g., Calculator, Settings). To further address the concerns on limited experiments we also added experiments on MuJoCo continuous control tasks (results reported in the table below)..
> >
> > **W5**. GAIL has extremely poor performance with only 10 trajectories, which is why we added more. Regardless, we will add extra results for BabyAI for GAIL in the low data regime. For MuJoCo, we evaluated GRACE and GAIL on the same number of trajectories.
> >
> > **Q4**. AndroidWorld is an emulator running the full Android OS. The reward functions generated by GRACE rely on the Android Accessibility Service layer (XML view hierarchy). Since this XML stream is a standard feature available on physical Android devices, the generated reward code is directly deployable to real devices without modification. We are not sure we understand the question about cross-app generalisation. The rewards are task-dependent (e.g., the framework can only learn how to start a timer from a trajectory starting a timer, not how to create an event in the calendar), however the reward structure generalizes. Because GRACE operates on the standard Android XML schema, the helper functions it discovers (e.g., logic to identify clickable buttons or parse time strings) are applicable across any Android application. We agree with the reviewer it would be interesting to test our rewards on similar apps (like a different Clock app like ClockBuddy), but we haven’t done so as this isn’t supported by AndroidWorld.
> >
> > **Q5**. We did not test the model with partial observability, but we did test it on raw pixels. We told the model it could use any libraries or open weights model it wanted, but unfortunately the high error rate of OCR / image models meant the model did not manage to write a reliable reward function (as if the feature extraction failed, which is often did, the code would be unable to recover from it). If a symbolic representation of the data is not available, we believe code is simply not the best solution for the problem, as we discuss in Appendix A4. We have moved this discussion to the Limitations section in the main body of the paper.

---

### Author Response · Authors · 2025-11-23

We thank the reviewers for their insightful feedback and constructive criticism. We feel encouraged that reviewers found the approach "novel" (R5gj), "interesting and promising" (2iny), and capable of "surpassing neural IRL methods in sample efficiency" (ZUCH). Reviewer JQHA further highlighted that our code-based representation enables “interpretability and natural emergence of reusable reward APIs."
Their comments have helped us to significantly strengthen the paper, pushing us to add new experiments on continuous control domains (MuJoCo) and open-weight models. Below, we summarize our response to the major themes raised across reviews.

## Core Contribution: Interpretable IRL vs. Reward Design
A recurring theme (Reviewers 2iny, R5gj, JQHA) was related to comparisons to methods like Eureka, Code-as-Reward, or LLM-as-Judge. We want to clarify the difference regarding our problem setting:
- Methods like Eureka assume access to a ground-truth reward function (or human feedback) to evaluate the fitness of generated code. GRACE, however, is an Inverse Reinforcement Learning (IRL) framework. It assumes no ground truth reward or reward description is available, it must recover the reward structure from expert demonstrations only. We could not compare against it because Eureka solves a different problem.
- Regarding VLM-Code-as-Reward: While this method also utilizes expert trajectories for verification, it relies on a VLM-specific pipeline designed for visual object detection (e.g., "identify red cubes") and explicitly acknowledges requiring manual assembly of the generated scripts [Venuto et al., Sec 5.2, 6]. In contrast, GRACE is a fully automated, end-to-end IRL framework that operates on state/symbolic representations without human-in-the-loop code stitching. VLM-CaR's prompting pipeline is incompatible and would require a significant redesign to work on your data modalities.
- Our Contribution: Our goal is to demonstrate that code generation is a viable, interpretable alternative to traditional black-box neural IRL (like GAIL). While GRACE currently relies on reasonably structured states (a limitation we acknowledge) we demonstrate it offers significant improvements in sample efficiency, debugging, and generalization for many control tasks.

## New experiments: Continuous Control MuJoCo tasks
Reviewers asked for evaluation on standardized continuous control benchmarks (2iny, JQHA), tests on robustness regarding the Goal Identification module (ZUCH, R5gj), and open-weights analysis (ZUCH). We have added a comprehensive series of experiments on MuJoCo (Hopper, Walker, Ant, Humanoid) to address this. Our new experiments demonstrate:
- GRACE successfully recovers rewards for continuous state spaces, refuting the concern that it is limited to discrete logic tasks.
- We ran all these new experiments without the Goal Identification module (in Phase 1 and 3), treating all expert states as positive and learner states as negative. GRACE successfully recovered effective rewards, proving the framework is robust and does not rely on explicit goal labeling to function.
- We evaluated GRACE using Qwen3-Coder-30B. It performed comparably to GPT-4o on MuJoCo tasks.

## Computational Cost Concerns
Reviewers 2iny, ZUCH, and JQHA queried the computational overhead. We clarify that GRACE is arguably more efficient than neural IRL methods in the standard RL loop:
- The initial evolutionary search is a one-time offline cost (approx. 1.00-10.00 USD per task via API, or minutes of GPU time via open weights) and generally only requires a few minutes
- Once the Python reward function is generated, it is orders of magnitude faster to query during the millions of timesteps in RL training compared to GAIL. GAIL requires a forward (and occasionally backward if the model needs to be updated) pass of a deep neural network discriminator at every timestep, whereas GRACE executes native Python code.
- GRACE does not require retraining the policy for every reward candidate in the population. Fitness is evaluated via static classification (Eq. 2). The policy is only trained once (or extremely sparsely) in Phase 3. We apologise that this was unclear in the text.

With the inclusion of the new MuJoCo experiments, open-model validations, and cost analyses, we believe GRACE stands as a “proof of existence” for interpretable, sample-efficient IRL method that outperforms standard baselines like GAIL in low-data regimes. We thank the reviewers for pushing us to demonstrate the generality of our method beyond the initial benchmarks and hope we have addressed all of their concerns.
We have also updated the submission PDF to include the new MuJoCo results, highlight the optionality of goal identification during Phase 1 of GRACE, and clarify the fitness normalization details.

---

> ### Author Response · Authors · 2025-11-26
>
> | Task | PPO | GRACE w/ GPT-4o | GRACE w/ Qwen3-Coder-30B | GAIL w/ 10 traj | GAIL w/ 200 traj |
> | :--- | :---: | :---: | :---: | :---: | :---: |
> | **Hopper** | $2212 \pm 54$ | $2143 \pm 80$ | $2106 \pm 76$ | $1902 \pm 183$ | $2056 \pm 92$ |
> | **Walker** | $2675 \pm 292$ | $2072 \pm 576$ | $2229 \pm 600$ | $790 \pm 90$ | $1982 \pm 101$ |
> | **Ant** | $6239 \pm 237$ | $5707 \pm 210$ | $6085 \pm 804$ | $3871 \pm 408$ | $5521 \pm 674$ |
> | **Humanoid** | $6455 \pm 302$ | $5809 \pm 106$ | $5921 \pm 301$ | $4772 \pm 251$ | $6521 \pm 337$ |
>
> **MuJoCo Results: Average returns on 4 MuJoCo (BRAX) continuous control tasks.** Average and standard deviation is reported across 5 different seeds. GRACE is always trained with 10 expert trajectories. The total number of required LLM calls to recover a reward for each task averages at 200 for both GPT-4o and Qwen3-Coder-30B.

---

### Meta-Review · Area_Chair_nFpM · 2026-01-02

**Summary:**

This paper proposes GRACE, an IRL framework that uses code generation LMs and an evolutionary search procedure to recover interpretable reward functions from expert demonstrations. Instead of learning black-box neural rewards, GRACE represents rewards as executable Python code and refines them using a fitness objective based on distinguishing expert from non-expert states. The paper evaluates the approach on BabyAI and AndroidWorld, showing strong sample efficiency and competitive downstream RL performance compared to GAIL, and demonstrates emergent modular reward structure across tasks.

Across reviews, the main concerns focused on (i) comparisons to prior LLM-based reward or code-as-reward methods (e.g. Eureka), (ii) computational cost and scalability, (iii) reliance on LLM-based goal identification, and (iv) the scope of experimental evaluation. The rebuttal and revised draft addressed many of these points with clarifications and additional experiments, particularly around computational cost, robustness of the framework without goal labeling, and generality to continuous control domains.

**Reviewer Concerns:**

The rebuttal appropriately addressed the following concerns:
- The rebuttal clarified that reward evolution is evaluated via static classification rather than RL-in-the-loop, and provided concrete cost estimates showing that reward synthesis is a one-time offline cost (on the order of dollars per task), after which the learned reward is cheap to evaluate during RL.
- Multiple reviewers worried that LLM goal labeling was a brittle or non-standard IRL assumption. The authors clarified that this module is optional and demonstrated additional experiments (MuJoCo continuous control) where GRACE operates without any goal identification, alleviating concerns that the method fundamentally depends on this component.
- The authors added results on multiple MuJoCo continuous control tasks, addressing concerns that the method only applies to discrete or logic-heavy domains.
- The rebuttal included results with an open-weights code model (Qwen3-Coder-30B), showing comparable performance to GPT-4-class models on added benchmarks.

However, there remain the following limitations:
- The conceptual distance between GRACE and prior LLM-based reward synthesis or code-as-reward approaches is still unconvincing, making the contribution slightly incremental within a rapidly growing design space. Authors might want to discuss and contrast the method to other methods like VoxPoser (https://voxposer.github.io/) or ALGAE (https://arxiv.org/abs/2409.08212).
- The evaluation, while broadened, still focuses on environments with structured state representations; applicability to raw sensory inputs remains limited and is acknowledged as such.

Overall, while not all conceptual disagreements were fully resolved, the core technical misunderstandings and feasibility concerns raised in the reviews were largely addressed.

**Reviewer Scores:**

- ZUCH would be likely to increase slightly from 4 to 6, as concerns about cost, robustness to goal labeling, and continuous control were directly addressed.
- R5gj would also likely increase their score possibly to a 4, as requested clarifications and additional experiments were provided.
- JQHA would likely keep their 6, with several major concerns explicitly addressed in the revision.
- 2iny would likely keep their 2, as their objections were more conceptual (novelty and positioning relative to prior work) rather than purely empirical. It's possible they'd be convinced by the authors' argument of comparison to EUREKA and increase their score to a 4.

---

### Decision · Program_Chairs · 2026-01-26

Accept (Poster)